# Mechanically induced correlated errors on superconducting qubits with relaxation times exceeding 0.4 ms

Shingo Kono [1,2,3] ✉, Jiahe Pan[1,2,3], Mahdi Chegnizadeh[1,2,3], Xuxin Wang[1,2], Amir Youssefi[1,2], Marco Scigliuzzo [1,2] & Tobias J. Kippenberg [1,2] ✉

Superconducting qubits are among the most advanced candidates for achieving fault-tolerant quantum computing. Despite recent significant advancements in the qubit lifetimes, the origin of the loss mechanism for state-of-the-art qubits is still subject to investigation. Furthermore, the successful implementation of quantum error correction requires negligible correlated errors between qubits. Here, we realize long-lived superconducting transmon qubits that exhibit fluctuating lifetimes, averaging 0.2 ms and exceeding 0.4 ms – corresponding to quality factors above 5 million and 10 million, respectively. We then investigate their dominant error mechanism. By introducing novel time-resolved error measurements that are synchronized with the operation of the pulse tube cooler in a dilution refrigerator, we find that mechanical vibrations from the pulse tube induce nonequilibrium dynamics in highly coherent qubits, leading to their correlated bit-flip errors. Our findings not only deepen our understanding of the qubit error mechanisms but also provide valuable insights into potential error-mitigation strategies for achieving fault tolerance by decoupling superconducting qubits from their mechanical environments.

Superconducting qubits have become a viable platform both for scientific and technological applications, ranging from fundamental quantum optical experiments[1], hybrid quantum systems[2] to quantum information science[3]. In particular, they have attracted much attention in fault-tolerant quantum computing, achieving important milestones, including the realization of high-fidelity quantum gate and readout on a multiple-qubit system[4,5], the reports of quantum supremacy[6,7], and the demonstrations of surface codes[8–10]. Despite such encouraging progress, realizing large-scale superconducting quantum computing is still an outstanding challenge[11]. Although quantum error correction promises reliable and scalable quantum computing, it strictly requires the physical errors of a large number of qubits to be sufficiently smaller than a certain threshold, and, more importantly, to be uncorrelated[12,13].

Since the physical errors in superconducting qubits are dominantly limited by their coherence[4,5,14], considerable efforts towards improvements in the qubit lifetimes have been made using insights from diverse fields[15], ranging from classical and quantum circuit engineering[16–20] to material science[21–25]. However, it still remains unclear whether the qubit lifetimes can be enhanced steadily. In addition, more coherent superconducting qubits are more sensitive to small changes in their environments[25], imposing a challenge on their scalability. For instance, when a superconducting qubit is dominantly coupled to a few two-level systems (TLSs), its relaxation time is largely fluctuating[26,27]. More recently, it has been reported that the absorption of ionizing radiation generates high-energy phonons in a qubit substrate, which leads to nonequilibrium quasiparticles, causing correlated charge-parity switching[28] and energy relaxations[29] of

[1]Institute of Physics, Swiss Federal Institute of Technology Lausanne (EPFL), Lausanne, Switzerland. [2]Center for Quantum Science and Engineering, EPFL, Lausanne, Switzerland. [3]These authors contributed equally: Shingo Kono, Jiahe Pan, Mahdi Chegnizadeh. ✉e-mail: shingo.kono@epfl.ch; tobias.kippenberg@epfl.ch

superconducting qubits. To verify fault tolerance, it is, therefore, more and more important to characterize a highly coherent multiple-qubit system more carefully, i.e., not only reporting their averaged coherence times but also studying the time and frequency dependence, as well as confirming the absence of correlated errors. Indeed, such characterizations have revealed dominant loss mechanisms and sources of fluctuations in superconducting qubits, such as surface dielectric loss[30], TLSs[26,31–33], nonequilibrium quasiparticles[34,35], and ionizing radiation[36–38].

Here, we realize long-lived superconducting transmon qubits based on niobium capacitor electrodes with the average and the longest lifetimes exceeding 0.2 ms and 0.4 ms, respectively, and report a new source of correlated bit-flip errors, caused by mechanical bursts generated by the pulse tube cooler of a dry dilution refrigerator[39]. This is revealed by a novel time-resolved analysis of the residual excited-state probabilities and quantum jumps of multiple qubits, which are synchronized with the operation of the pulse tube cooler. Although the origin of the mechanical sensitivity of long-lived superconducting qubits could not be determined unambiguously in this work, our observations are consistent with quasiparticle- and TLS-mediated qubit decays to phononic baths[40,41]. Moreover, these findings suggest future strategies for fault-tolerant superconducting quantum computing, including the development of acoustically shielded superconducting devices[42–44], mechanical shock-resilient sample packaging[45,46], and a vibration-free dilution refrigerator[47–49].

## Results

### Long-lived superconducting transmon qubits

We develop superconducting transmon qubits, each formed by a single $Al/AlO_x/Al$ Josephson junction shunted by a Nb capacitor, fabricated on a high-resistivity silicon substrate (see the fabrication details in Supplementary Note 1). Figure 1a, c show an optical micrograph of a

fabricated multiple superconducting qubit device and its equivalent circuit model, respectively, including four frequency-fixed transmon qubits ($Q_0$–$Q_3$) with resonance frequencies ranging from 4.8 GHz to 6.2 GHz and anharmonicities of −0.26 GHz on average. As the metal-substrate interface of the Al film fabricated by a lift-off process can not be as clean as that of the Nb film directly sputtered on the silicon substrate, we minimize the area of the Al electrodes and bandage patches[50] to reduce the energy participation ratio in the interface (see Fig. 1b). To realize multiplexed dispersive readout, all the qubits are individually coupled to $\lambda/4$ readout resonators with different resonance frequencies around 7 GHz, sharing a $\lambda/2$ Purcell filter[51]. The filter is connected to a feed line, along which frequency-multiplexed control and readout signals are sent. The filter is designed to have a 7 GHz resonance frequency and a 300 MHz bandwidth, suppressing the qubit radiative decay rates to a level of $\mathcal{O}(10\,\text{Hz})$. The state-dependent dispersive shifts and the readout resonator bandwidths are designed to be $\mathcal{O}(1\,\text{MHz})$. Note that the different dispersive shifts of the readout resonators for the first and second excited states enable us to distinguish between the readout signals corresponding to the first three states ($G$, $E$, and $F$) in a single shot. See Table 1 in Method for the full characterization of the system parameters.

As schematically shown in Fig. 1d, the fabricated device is mounted at the mixing chamber stage (-10 mK) of a dry dilution refrigerator, enclosed in a multilayer shielding: copper radiation shields and magnetic shields of aluminum and mu-metal. The transmon qubits are characterized using a nearly quantum-limited broadband Josephson traveling wave parametric amplifier (JTWPA)[52], allowing us to perform the simultaneous single-shot readout of the qubits by frequency-multiplexing[5]. The readout errors of the $G$ states, primarily determined by separation errors, yield $\lesssim 0.08\%$ for $Q_0$ and $\lesssim 0.04\%$ for $Q_1$, while those of the $E$ ($F$) states, primarily influenced by state-flip errors, yield $\lesssim 2\%$ (5%) for $Q_0$ and $\lesssim 5\%$ (8%) with additional influence from separation errors for $Q_1$ (see more

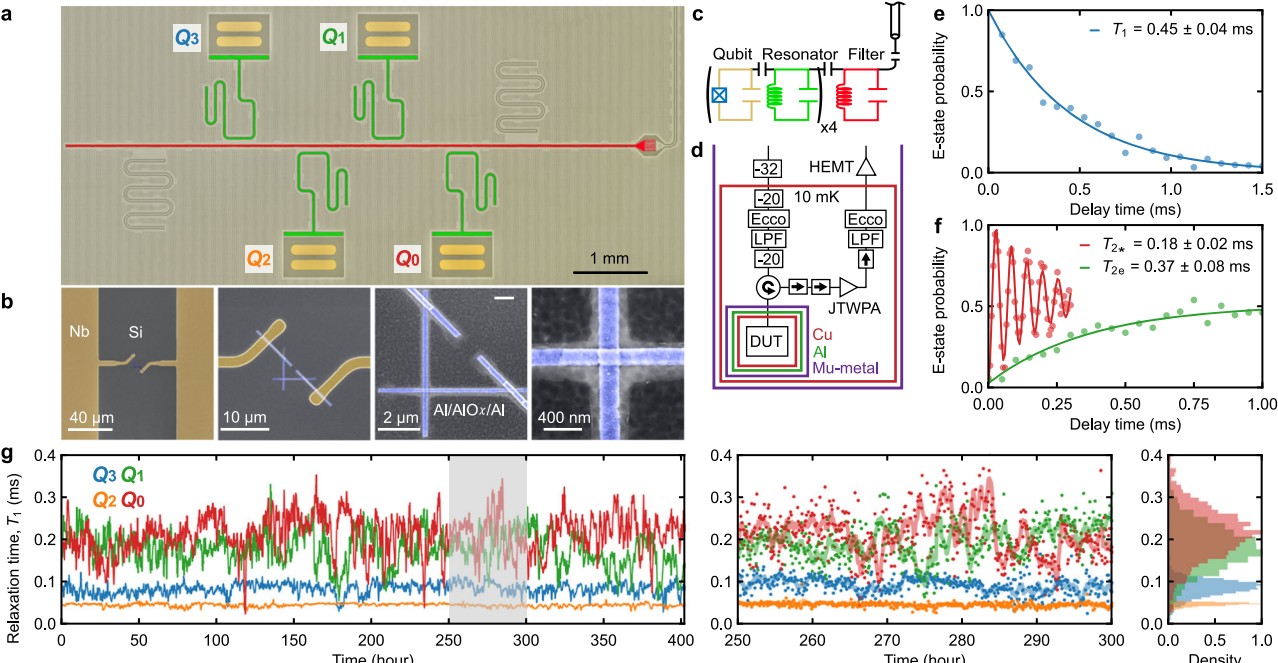

**Fig. 1 | Long-lived superconducting transmon qubit device based on Nb capacitor electrodes. a** False-colored optical micrograph of a superconducting device, consisting of four transmon qubits (yellow, $Q_0$–$Q_3$) with individual readout resonators (green) coupled to a shared Purcell filter (red). **b** False-colored scanning electron microscope images of an $Al/AlO_x/Al$ Josephson junction (blue) shunted by a Nb capacitor (yellow) on a silicon substrate (gray). **c** Equivalent circuit of the device. **d** Simplified cryogenic wiring. **e, f** Time traces of the excited-state

probability of qubit $Q_0$, showing the longest measured values of the relaxation time and the Ramsey and Hahn-echo dephasing times. **g** Long-term stability of the relaxation times of the four qubits. The middle panel shows the magnified plot for the gray region of the left panel, where the dots are the results obtained from the individual time traces, while the solid lines are their smoothed data with a 5 h time window. The right panel shows the height-wise normalized histograms.

**Table 1 | System parameters**

| Parameter | $Q_0$ | $Q_1$ | $Q_2$ | $Q_3$ |
|---|---|---|---|---|
| Qubit frequency, $\omega_q/2\pi$ (GHz) | $4.794064 \pm 8 \times 10^{-6}$ | $5.20603 \pm 2 \times 10^{-5}$ | $5.721 \pm 1 \times 10^{-4}$ | $6.23127 \pm 3 \times 10^{-5}$ |
| Anharmonicity, $\alpha/2\pi$ (GHz) | −0.272 | −0.266 | −0.263 | −0.250 |
| Relaxation time of $E$, $T_1$ (ms) | $0.21 \pm 0.06$ | $0.18 \pm 0.05$ | $0.04 \pm 0.005$ | $0.08 \pm 0.02$ |
| Ramsey dephasing time of $G$-$E$, $T_{2^*}$ (ms) | $0.1 \pm 0.05$ | $0.06 \pm 0.03$ | $0.02 \pm 0.01$ | $0.06 \pm 0.03$ |
| Echo dephasing time of $G$-$E$, $T_{2e}$ (ms) | $0.29 \pm 0.09$ | $0.22 \pm 0.07$ | $0.08 \pm 0.01$ | $0.08 \pm 0.03$ |
| Relaxation time of $F$, $T_{1F}$ (ms) | $0.1 \pm 0.02$ | $0.08 \pm 0.01$ | - | - |
| Simulated Purcell limit (ms) | 127 | 35 | 26 | 0.2 |
| Resonator frequency, $\omega_c/2\pi$ (GHz) | 7.07605 | 6.97984 | 6.885998 | 6.797376 |
| External coupling rate, $\kappa_{ex}/2\pi$ (MHz) | 1.85 | 1.06 | 0.102 | 0.52 |
| Internal loss rate, $\kappa_{in}/2\pi$ (MHz) | 0.11 | <0.01 | 0.02 | 0.01 |
| Dispersive shift for $E$, $\chi_{GE}/2\pi$ (MHz) | 0.65 | 0.95 | 1.7 | 6.6 |
| Dispersive shift for $F$, $\chi_{GF}/2\pi$ (MHz) | 1.04 | 1.55 | - | - |
| Readout cavity photon number for $G$ | $2.7 \pm 0.9$ | $0.7 \pm 0.3$ | - | - |
| Readout cavity photon number for $E$ | $5 \pm 2$ | $3 \pm 1$ | - | - |
| Readout cavity photon number for $F$ | $6 \pm 2$ | $2 \pm 1$ | - | - |

The dispersive shifts, denoted by $\chi_{GE}$ ($\chi_{GF}$), are the frequency difference between the resonators with the qubits in the $G$ and $E$ ($F$) states.

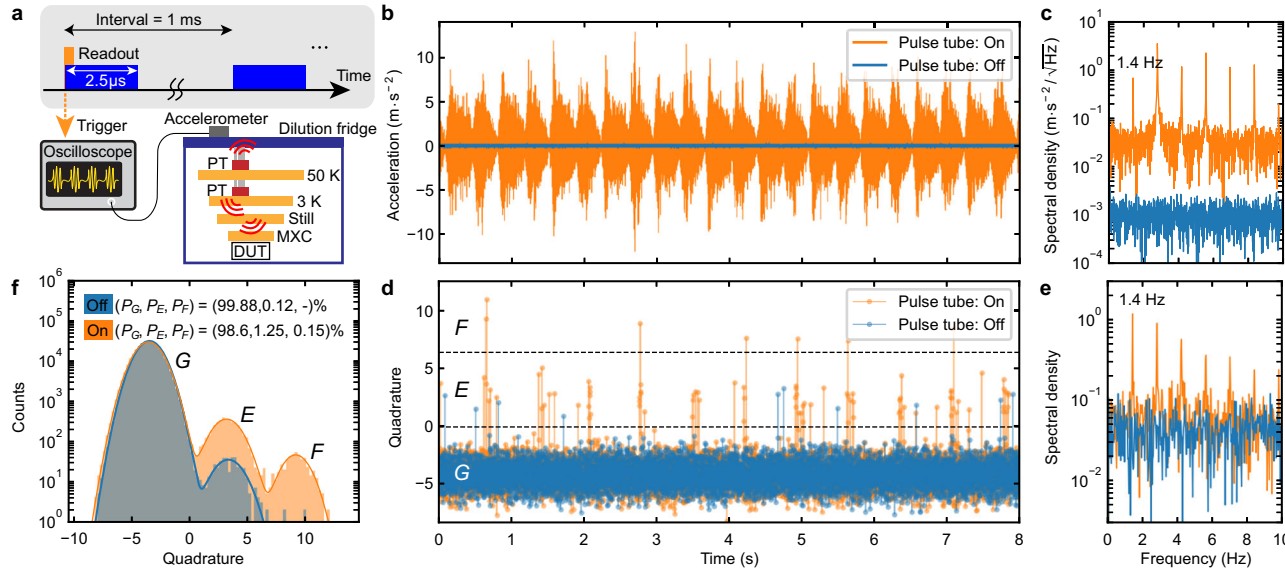

**Fig. 2 | Out of equilibrium dynamics of a transmon qubit, induced by mechanical vibrations from the pulse tube cooler. a** Pulse scheme and simplified experimental setup for simultaneously recording both the vibrational noise generated by the pulse tube cooler (PT) and the single-shot qubit readout outcomes. **b** Time traces of the acceleration of the vibrational noise with the pulse tube cooler on (orange) and off (blue), and **c** the amplitude spectral densities (ASD) of their absolute values. The fundamental frequency of the harmonics is approximately 1.4 Hz. **d** Time traces of the readout quadrature amplitudes and **e** their ASDs.

The black dashed lines show a threshold to distinguish between the $G$ and $E$ states and one between the $E$ and $F$ states, respectively. The quadrature amplitudes are normalized by the $G$-state quadrature standard deviation. The same normalization is carried out in every single-shot data in this work. **f** Histograms of the qubit readout quadrature amplitudes when the pulse tube cooler is on and off. The solid lines are the mixture of Gaussian distributions fitted to the histograms in order to obtain the occupation probabilities.

details in Supplementary Note 3). To suppress thermal and backward amplifier noises, the input and output lines are heavily attenuated and isolated, respectively, while both are equipped with low-pass filters and eccosorb filters (see more details in "Methods" section).

Figure 1e, f show the time traces of the excited-state probability of the transmon qubit with a resonance frequency of 4.8 GHz ($Q_0$), showing the longest relaxation times ($T_1 = 0.45 \pm 0.04$ ms) and the longest Ramsey and Hahn-echo dephasing times ($T_{2^*} = 0.18 \pm 0.02$ ms and $T_{2e} = 0.37 \pm 0.08$ ms), respectively. Our observations confirm that coherence improvements are still possible with widely employed superconducting material systems involving Al/AlOx/Al Josephson junctions and Nb electrodes fabricated on a silicon substrate[20].

As shown in Fig. 1g, we measure the long-term stability of the relaxation times of the four qubits, showing significantly large fluctuations, especially for the longer-lived qubits ($Q_0$ and $Q_1$) with average $T_1$ of approximately 0.2 ms and relative standard deviations of 30%, while $Q_2$ and $Q_3$ with average $T_1$ of 0.04–0.08 ms exhibit smaller relative deviations of 10–20%. The Allan deviation analysis of the fluctuations implies that the relaxation times of our long-lived transmon qubits are mainly limited by TLSs[27] (see Supplementary Note 2).

### Effect of pulse tube cooler on qubit excitations

We perform the single-shot readout for the longest lifetime transmon qubit ($Q_0$). As shown in Fig. 2a, we apply 2.5 μs long readout pulses

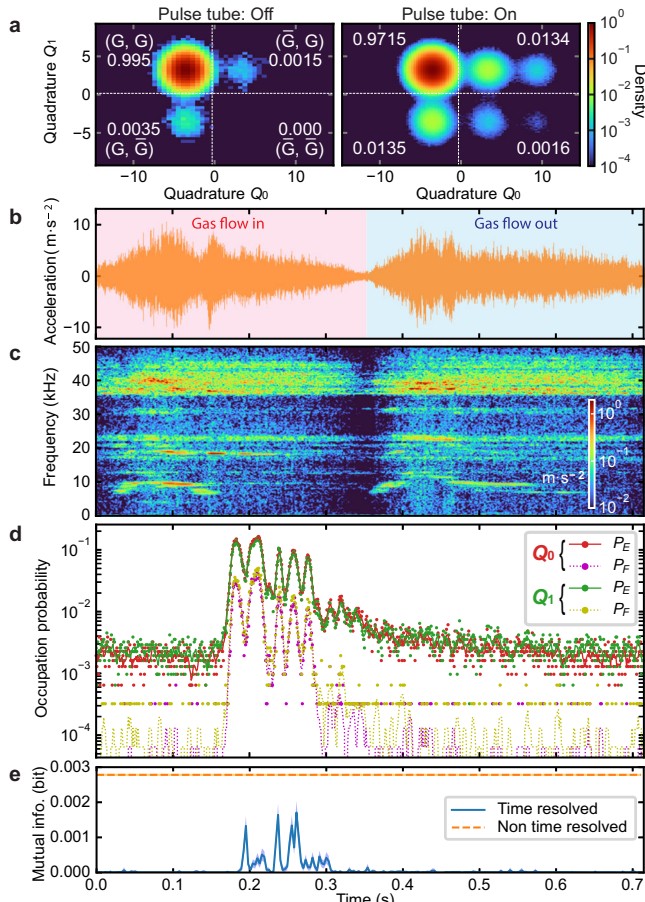

**Fig. 3 | Mechanically induced correlated qubit excitations. a** Height-wise normalized 2D histograms of the simultaneous readout quadrature outcomes for qubits $Q_0$ and $Q_1$ with the pulse tube cooler off and on, where the number of data points are approximately $3 \times 10^5$ and $10^7$, respectively. **b** Time trace of the acceleration of a referential pulse tube vibrational noise within one period and **c** the result of its time-frequency analysis. There are two phases in one cycle of the pulse tube operation: helium gas flow in (red shaded) and out (blue shaded). **d** Time-resolved $E$- and $F$-state probabilities for $Q_0$ and $Q_1$, time-aligned to the referential vibrational noise. The dots and lines are the raw data with a 1 ms time resolution and their smoothed data with a 5 ms time window, respectively. **e** Mutual information as a function of time within the one vibrational period, obtained from the time-resolved simultaneous single-shot data for $Q_0$ and $Q_1$. The blue solid line is the time-resolved MI, while the orange dashed line is the non-time-resolved one. The confidence interval of one standard error is shown with the shaded region.

repeatedly with an interval of 1 ms ($\approx 5$ times larger than the average $T_1$), which is sufficiently long to prepare the qubit in the equilibrium. In addition to the conventional qubit measurement setup, we anchor an accelerometer on the top plate of the dilution refrigerator, converting the pulse tube vibrational noise to a voltage signal. The converted signal is acquired by an oscilloscope that is operated synchronously with the qubit readout sequence via a trigger signal generated by the qubit measurement setup, enabling us to simultaneously record both the vibrational noise and the single-shot readout outcomes.

Figure 2b, d show the synchronized time trace records of the vibrational noise and the qubit single-shot readout quadrature amplitudes, respectively. When the pulse tube cooler is on (orange), the qubit is more frequently excited to the first excited state ($E$), even to the second excited state ($F$), while with the cooler off (blue), it mostly remains in the ground state ($G$). Here, we obtain the data with the pulse tube cooler off by deactivating it temporarily without affecting the base temperature ($\approx 5$ min). Note that the readout signals for the $E$ and $F$ states of $Q_0$ can be distinguished well in one quadrature projected in

an optimal phase. Importantly, when the pulse tube cooler is on, the qubit becomes excited periodically in time, synchronized with the periodic vibrational noise. Figure 2c, e show the amplitude spectral densities of the vibrational noise and the qubit readout quadrature, respectively, showing both have harmonics with exactly the same fundamental frequency of approximately 1.4 Hz when the pulse tube cooler is on. Note that we show the spectral density of the absolute value of the raw vibrational data to capture the pulse tube repetition frequency (see Supplementary Note 7).

Figure 2f shows the histograms of the readout quadratures (the number of each data $\approx 3 \times 10^5$) with the pulse tube cooler on and off. To mitigate the readout separation errors, we obtain the occupation probability in each qubit state by fitting the histogram to a mixture of multiple Gaussian distributions. When the pulse tube cooler is switched off, we achieve a high initialization fidelity of 99.88% ($P_E = 0.12\%$) by passive cooling, where the corresponding effective temperature is $T_{\mathrm{eff}} = 34$ mK, although the base temperature is approximately 10 mK. Importantly note that the excited-state probability due to the state-flip error induced by the readout backaction is $\lesssim 0.02\%$, not dominantly limiting the residual qubit excitation (see Supplementary Note 3). In contrast, when the pulse tube is on, the qubit is excited to the $E$ state with a higher probability ($P_E = 1.25\%$), and even the occupation probability in the $F$ state is not negligible ($P_F = 0.15\%$). More interestingly, the occupation probability distribution is not in the thermal equilibrium, i.e., the effective temperature is $T_{\mathrm{eff}} = 53$ mK for the $G$-$E$ transition, while $T_{\mathrm{eff}} = 102$ mK for the $E$-$F$ transition. This implies that the qubit is not simply excited by a local thermal heating of the environment, but it is excited by nonequilibrium dynamics of the mechanical vibrations generated by the pulse tube cooler.

## Mechanically induced correlated excitations

We study the existence of correlated excitations for two of the long-lived qubits, $Q_0$ and $Q_1$ with the average $T_1 \approx 0.2$ ms, by simultaneously reading out both states with a frequency-multiplexed readout pulse. Figure 3a shows the 2D histograms of the simultaneous readout outcomes of approximately $3 \times 10^5$ and $10^7$ measurements when the pulse tube cooler is off and on, respectively. Note that the different numbers of data points are due to the limited measurement time when the pulse tube is switched off. As shown in Fig. 3a, we can obtain the excited-state probability for both qubits, or the probability when the state is found in the $E$ or $F$ states ($\bar{G}$), by using only the real part of the readout complex amplitude. To quantitatively study the correlated excitations, we use mutual information (MI) in the unit of bit, which is defined as

$$I = \sum_{\mathcal{X},\mathcal{Y}} P(\mathcal{X},\mathcal{Y}) \log_2 \left[ \frac{P(\mathcal{X},\mathcal{Y})}{P(\mathcal{X}) \cdot P(\mathcal{Y})} \right]. \quad (1)$$

Here, $P(\mathcal{X},\mathcal{Y})$ is the joint probability of qubit $Q_0$ in the $\mathcal{X}$ state and qubit $Q_1$ in the $\mathcal{Y}$ state, while $P(\mathcal{X})$ and $P(\mathcal{Y})$ are the marginal probabilities of $Q_0$ in $\mathcal{X}$ and $Q_1$ in $\mathcal{Y}$, respectively, where $\mathcal{X}$ and $\mathcal{Y}$ can be $G$ and $\bar{G} = E$ or $F$. This quantifies how much information about the excitation of one qubit we obtain from the other qubit ($0 \leq I \leq 1$ bit). We correct the systematic bias and obtain the standard error on the estimated MI with the method described in ref. 53. For the case when the pulse tube cooler is off, the MI is $I < 10^{-6}$ bit, while that with the pulse tube on is found to be $I = 0.00278$ bit (orange dashed line in Fig. 3e), showing there is a significant correlation in their excitations that are induced by the pulse tube mechanical vibrations.

To investigate the origin of the correlated excitations, we develop a time-resolved analysis of the qubit excited-state probabilities, synchronized with the periodic vibrational noise generated by the pulse tube cooler. Using the protocol shown in Fig. 2a, we repeat a sequence consisting of approximately 4000 multiplexed readout pulses ($\approx 4$ s in total) while simultaneously recording the vibrational noise. Since the starting time of each sequence is not synchronized with the phase of

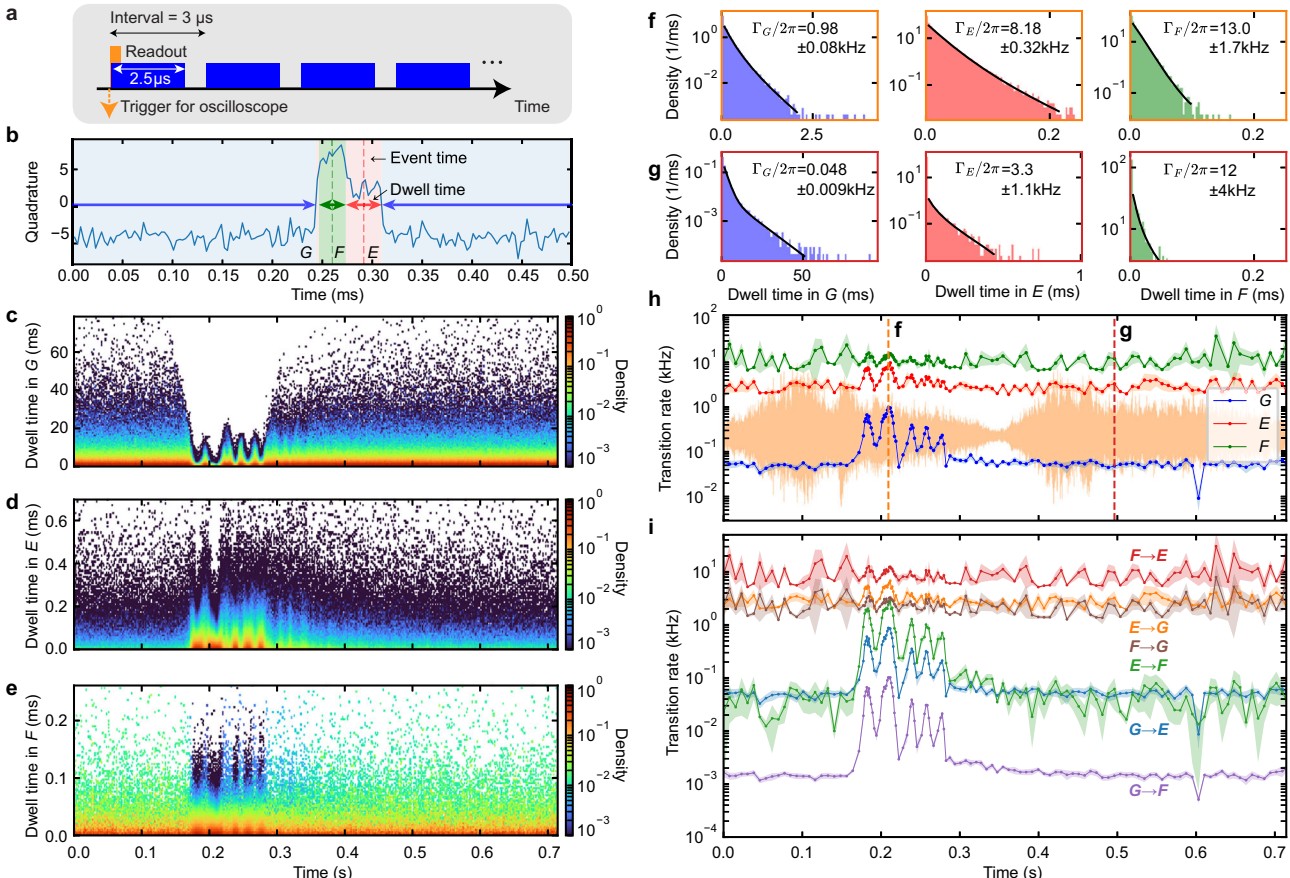

**Fig. 4 | Mechanically induced quantum jumps. a** Pulse scheme for continuously monitoring the states of qubits $Q_0$ and $Q_1$, where multiplexed readout pulses are repeated successively, synchronized with the measurement of the vibrational noise from the pulse tube cooler. **b** An example of a time trace of the single-shot readout quadrature for $Q_0$, showing several quantum jumps. The shaded blue, red, and green regions correspond to the dwell events in the $G$, $E$, and $F$ states, respectively. The event time, centered at each dwell event, is shown with a vertical dashed line, while the dwell time is shown with a double-sided horizontal arrow. **c**–**e** 2D histograms of the event time and the dwell times in the $G$, $E$, and $F$ states, respectively. The histograms are normalized by the height at the first dwell time bin for every event time. **f, g** Time-resolved dwell-time distributions for $G$, $E$, and $F$, and for different times specified with dashed lines in (**h**). The black lines are the fitting results. **h** Time-resolved total transition rates from the $G$, $E$, and $F$ states to the other two ($\Gamma_G$, $\Gamma_E$, and $\Gamma_F$), respectively, time-aligned to the referential periodic vibrational noise (orange). **i** Time-resolved individual transition rates among the $G$, $E$, and $F$ states. The shaded regions in (**h**, **i**) depict the errors obtained from the fitting errors of the dwell-time distributions and the statistical errors of the conditional probabilities.

the periodic vibrational noise, we need to time-align every time trace of the single-shot data with respect to the periodic vibrational noise. To this end, we first specify one period of the vibrational noise as a reference, as shown in Fig. 3b. Then, we time-align every trace of the single-shot outcomes by maximizing the cross-correlation of the simultaneously recorded vibrational noise with the referential one. Consequently, we can accumulate a sufficient number of the single-shot outcomes at an arbitrary time of interest within the vibrational period to obtain the time-resolved excited-state probabilities of the qubits. Figure 3d shows the results of the time-resolved measurements of the $E$- and $F$-state probabilities for both qubits, obtained from the sequences repeated approximately 3000 times. Although the two operational phases of the pulse tube cooler exhibit similar vibrational noises, the qubits are frequently excited only during the gas-flow-in phase, significantly deviating from thermal distributions. Figure 3c shows the result of the time-frequency analysis of the vibrational noise, by taking a short-time Fourier transform with a 5 ms Hann window, revealing there is a difference between the gas-flow-in and out phases. However, there is no clear correlation between the time-resolved excited-state probabilities and the result of the time-frequency analysis up to 50 kHz. This could be because superconducting qubits can be excited by giga-hertz or higher frequency vibrational noise that cannot be measured by the accelerometer as long as a multi-phonon

excitation process and local heating do not play a dominant role. Moreover, we find that the $E$- and $F$-state probabilities of both qubits are increased synchronously from their lowest values ($P_E \approx 0.2\%$ and $P_F \approx 0.005\%$) by a factor of approximately one hundred and one thousand, respectively. This implies that the two qubits are dominantly excited by the common mechanical vibrations via their phononic baths.

Figure 3e shows the time-resolved MI as a function of time within the one vibrational period, which is obtained from the time-resolved multiplexed single-shot readout outcomes for the two qubits using Eq. (1). The time-resolved MI (blue solid line) consistently exhibits smaller values compared to the non-time-resolved counterpart (orange dashed line). This implies that individual qubit excitation events are not strongly correlated, while the excited-state probabilities vary in time synchronously between the qubits due to the global effect of the pulse tube mechanical vibrations, explaining the existence of the correlation observed in the non-time-resolved data.

## Mechanically induced correlated quantum jumps

Next, we study the mechanical effect on quantum jumps of the transmon qubits, i.e., the pulse tube effect on their transition rates among the $G$, $E$, and $F$ states. As shown in Fig. 4a, we repeat a sequence to continuously monitor the states of $Q_0$ and $Q_1$ about 3000 times[54],

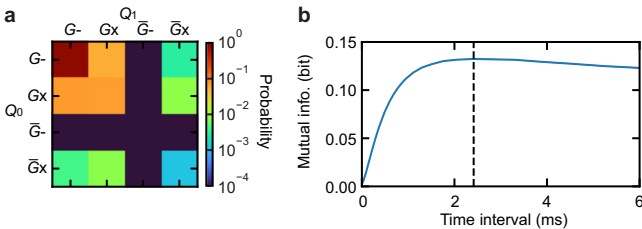

**Fig. 5 | Correlation analysis on quantum jumps. a** Table of the total 16 error probabilities of qubits $Q_0$ and $Q_1$, taken at the time interval which maximizes the MI, where the data is clipped to the minimum of $10^{-4}$. **b** Mutual information (MI) between the error probabilities of $Q_0$ and $Q_1$, as a function of the time interval. The dashed line marks the time interval showing the maximal MI.

while simultaneously recording the vibrational noise from the pulse tube cooler. Each sequence consists of approximately $5 \times 10^5$ successive 2.5 μs long multiplexed readout pulses with an interval of 3 μs, corresponding to a length of ≈1.5 s. Here, we will first focus on the time-resolved measurement of the transition rates of qubit $Q_0$ (Fig. 4), and then study the existence of a correlation in quantum jumps between $Q_0$ and $Q_1$ (Fig. 5).

In a similar manner to the previous time-resolved measurements, we can time-align every continuous monitoring trace with respect to the referential periodic vibrational noise, displayed in Fig. 3b. As shown in Fig. 4b, every time-aligned trace contains several quantum jump events, from which we sample the event time (vertical dashed line) and the dwell time (a double-sided horizontal arrow) for the $G$, $E$, and $F$ states, individually. Figure 4c–e show the 2D histograms of the event time and the dwell time for the three states, respectively, where each histogram contains >$10^6$ dwell events.

To determine the transition rates among the $G$, $E$, and $F$ states in a time-resolved fashion, we calculate the distribution of the dwell time in each state at a time of interest within the vibrational period. Namely, the time-resolved dwell-time distributions are obtained by using the dwell events within a bin centered at the chosen time. In addition, the bin width for all three states is set to be proportional to the average $G$ dwell time around the chosen time. This is because the time resolution of the measurement of the transition rate for the $G$ state is limited by the inverse of the rate, approximately corresponding to the average of the $G$ dwell time. Furthermore, the number of the $E$ ($F$) dwell events within a bin width of the average $E$ ($F$) dwell time (fundamental time resolution) is not sufficient, requiring a wider bin width to accumulate more data points, and eventually limiting the time resolution.

Figure 4f, g show examples of the time-resolved dwell-time distributions in the three states at the times specified with the dashed lines in Fig. 4h, respectively. As exemplified by the $G$ dwell-time distribution in Fig. 4g, some dwell-time distributions display two distinct characteristic timescales, for which an unambiguous reason is not identified in this study. However, similar phenomena are observed in various superconducting qubit systems, potentially linked to none-quilibrium quasiparticle dynamics[55–57]. To include this effect and accurately obtain the transition rates, we fit each dwell-time distribution to two distinct models: an exponential distribution and a mixture of two exponential distributions, and choose the fitting model with the lower Bayesian information criterion value (see more details in Supplementary Note 3). As shown in Fig. 4h, this analysis results in the time-resolved total transition rates from the $G$, $E$, and $F$ states to the other two ($\Gamma_G$, $\Gamma_E$, and $\Gamma_F$), respectively, as a function of time within one period of the vibrational noise.

To extract the individual transition rate from the $\mathcal{X}$ state to the $\mathcal{Y}$ state ($\Gamma_{\mathcal{X} \to \mathcal{Y}}$) based on the inferred total transition rate ($\Gamma_{\mathcal{X}} = \Gamma_{\mathcal{X} \to \mathcal{Y}} + \Gamma_{\mathcal{X} \to \mathcal{Z}}$), we utilize the probability of eventually transitioning to $\mathcal{Y}$ conditioned on dwelling in $\mathcal{X}$, where $\mathcal{X}$, $\mathcal{Y}$ and $\mathcal{Z}$ traverse the $G$, $E$, and $F$ states. Since the conditional probability of the event

yields $\Gamma_{\mathcal{X} \to \mathcal{Y}} / \Gamma_{\mathcal{X}}$, which can be experimentally inferred as the number of the $\mathcal{X}$ dwell events resulting in transitions to $\mathcal{Y}$ divided by the total number of the $\mathcal{X}$ dwell events, we can determine $\Gamma_{\mathcal{X} \to \mathcal{Y}}$ as the product of the total transition rate $\Gamma_{\mathcal{X}}$ and the conditional probability $\Gamma_{\mathcal{X} \to \mathcal{Y}} / \Gamma_{\mathcal{X}}$. Note that we correct a nontrivial effect of the readout separation errors on the inferred transition rates. See more details in Supplementary Note 4.

Figure 4i shows the time-resolved transition rates among the $G$, $E$, and $F$ states. The transition rates exhibit a dynamic pattern consistent with the previously described behavior observed in the excited-state populations (Fig. 3d). In detail, they show an interval where the pulse tube mechanical vibrations induce significant increases in all the transition rates, including the direct transition processes between the $G$ and $F$ states. Interestingly, a nonnegligible direct transition process between the $G$ and $F$ states is observed even in the "quiet" period, when the pulse tube effect is minimal. This can be explained by readout-induced higher-order transitions[58] or auxiliary mode-assisted two-photon transitions[59]. Finally, in the "quiet" period, the decay rates of the $E$ and $F$ states are larger compared to those in the free evolution (the inverse of the respective relaxation times). We attribute such increases to the readout-induced state flip[60].

Using the same continuous monitoring data, we further study whether there is a correlation in quantum jumps between qubits $Q_0$ and $Q_1$. When the qubits are continuously monitored for a certain time interval, an error, corresponding to a state transition, will occur with a finite probability. For simplicity, we do not distinguish between the $E$ and $F$ states in the correlation analysis ($\bar{G} = E$ or $F$). We chunk the continuous monitoring data and classify them into 16 possible events: the 2 possible initial states $\{G, \bar{G}\}$ and the error occurrence within the time interval $\{-, \times\}$ for each of the 2 qubits. Here, "−" stands for the occurrence of no bit-flip event, and "×" stands for the occurrence of at least one bit-flip event. As shown with an example in Fig. 5a, we can, therefore, calculate a $4 \times 4$ matrix of the error probabilities at a specific time interval.

To quantitatively study the existence of correlated bit-flip errors during continuous monitoring, we use mutual information (MI), defined in Eq. (1), using the $4 \times 4$ error probability matrix in this case. The MI corresponds to a quantitative value of the amount of the correlation in the quantum jumps, i.e., how much information about the occurrence of a bit-flip error in one qubit we can obtain from the other one ($0 \leq I \leq 2$ bit). Figure 5b presents the MI of the bit-flip error probabilities between $Q_0$ and $Q_1$ as a function of the time interval, showing the maximum of $I = 0.1328$ bit at 2.5 ms. It clearly shows that there is a correlation in the quantum jump events between the two qubits, which corresponds to the existence of a correlated error in their gates.

## Effect of a controlled mechanical shock

To rule out that the qubit excitations are due to possible electrical noise produced by the pulse tube cooler, we mount on the top plate of the dilution refrigerator an electric hammer based on an electromagnet (see Fig. 6a). In this manner, we generate a purely mechanical shock by a pulsed DC current in the electromagnet, synchronized with the qubit readout sequence via a trigger signal. We simultaneously record both the vibrational noise and the qubit single-shot readout outcomes while the pulse tube cooler is deactivated.

Figure 6b, c present the time traces of the acceleration generated by the controlled mechanical shock and the result of the time-frequency analysis, respectively, showing a broadband impulse shock, followed by a damped oscillation at eigen frequencies of the refrigerator. Figure 6d shows the time-resolved $E$- and $F$-state probabilities of qubits $Q_0$ and $Q_1$, measured synchronously with the mechanical shock. The excited-state probabilities are obtained from the single-shot outcomes of approximately $10^4$ measurements for each time of interest with a 10 ms time resolution. Note that we accumulate 1 ms time-resolved data sets for 10 ms to obtain a sufficiently large number

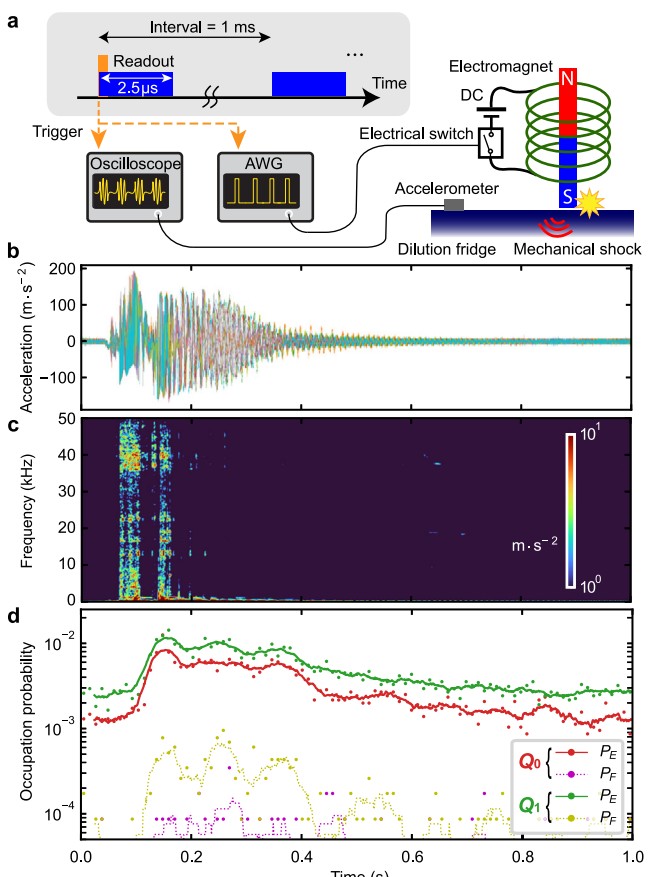

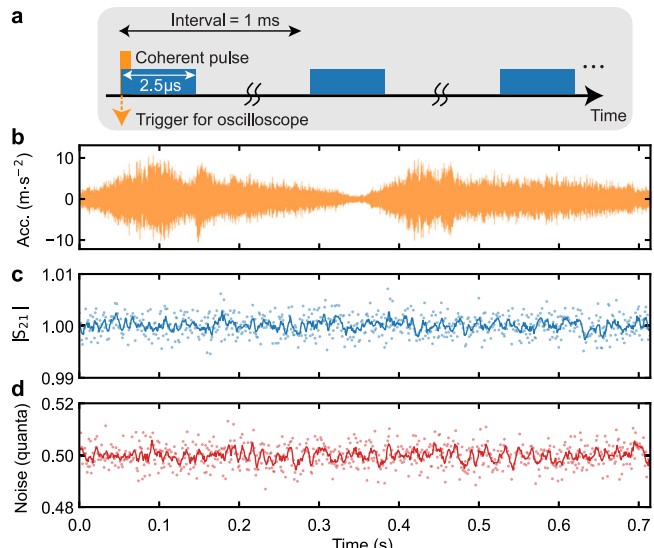

**Fig. 7 | Time-resolved measurement of the microwave bath. a** Pulse scheme for characterizing the qubit microwave bath, where 2.5 μs long coherent pulses are repeated with an interval of 1 ms, synchronized with the measurement of the vibrational noise from the pulse tube cooler. **b** Time trace of the acceleration of the pulse tube vibrational noise within one period. **c, d** Transmission coefficient and background photon noise of the microwave bath, measured in a time-resolved fashion and synchronized with the periodic vibrational noise. The transmission coefficient and background noise are normalized by the non-time-resolved values, respectively. Moreover, the background noise is re-scaled to the vacuum noise level (1/2), to be in the unit of quanta. The dots and lines are the raw data with a 1 ms time resolution and their smoothed data with a 5 ms time window, respectively.

**Fig. 6 | Qubit excitations induced by a controlled pure mechanical shock.**
**a** Pulse scheme and simplified experimental setup for simultaneously recording the qubit single-shot readout outcomes and a vibrational noise generated by a controlled mechanical shock. In addition to the oscilloscope setup for the vibration measurement, an AWG sends a synchronized switching signal to trigger a current in an electromagnet, leading to a mechanical shock. **b** Multiple time traces of the acceleration signal generated by the controlled mechanical shock and **c** the result of time-frequency analysis of one of them. **d** Time-resolved *E*- and *F*-state probabilities for qubits $Q_0$ and $Q_1$, synchronized with the controlled mechanical shock while the pulse tube cooler is deactivated. The dots and lines are the raw data with a 10 ms time resolution and their smoothed data with a 50 ms time window, respectively.

of data points to characterize the small *F*-state probabilities. We observe that both qubits are similarly excited purely by the mechanical shock, while its effect on the microwave bath is negligible (see Supplementary Note 6). This implies that possible electrical noise generated from the pulse tube cooler does not dominantly contribute to the qubit excitations. Interestingly, the qubits are not excited more frequently by the electric hammer when compared to the case of the pulse tube mechanical shock although the hammer mechanical shock is larger even when the mechanical shocks for the two cases are compared at the qubit device at room temperature (see Supplementary Note 7). We believe that this is because giga-hertz or higher frequency vibrational noise, which can not be characterized by the accelerometer and could be different between the two cases, needs to be responsible for the qubit excitations.

## Time-resolved measurement of microwave bath

One possible process to excite the qubits mechanically is that a side effect of a mechanical shock would change the property of the microwave bath, leading to an increase in the microwave background noise. For example, a change in the transmission through the filters

and attenuators may alter the amount of thermal noise coming from the higher temperature stages of the refrigerator. In addition, we use an attenuator and a terminator made of crystalline quartz in order to acoustically thermalize the feed line to the base temperature (see the detailed experimental setup in "Methods" section). Such microwave components would convert a mechanical shock to microwave noise, eventually exciting the qubits electrically. Moreover, local heating and a triboelectric effect could also increase the effective temperature of the microwave bath. These concerns motivate us to perform the time-resolved analysis of the microwave transmission coefficient ($|S_{21}|$) and background noise of the measurement chain, synchronized with the measurement of the periodic vibrational noise generated by the pulse tube cooler.

As shown in Fig. 7a, we apply coherent pulses at around 7 GHz, which are off-resonant with the readout resonators and qubits. By measuring the average and variance of the coherent amplitude of the pulses in a time-resolved manner (similar to the experiment for Fig. 3), we can characterize the microwave transmission coefficient and background noise at a time of interest within the period of the vibrational noise (Fig. 7b). Figure 7c, d show the time-resolved microwave transmission coefficient ($|S_{21}|$) and background photon noise, respectively, confirming that there is no significant time variation.

Here, we can estimate at least how much background noise increase is required to excite the qubits to the extent observed in Fig. 3d. First, we consider the scenario that the qubits are excited by mechanically induced microwave noise in the feedline. From numerical simulations based on the finite-element method, the external coupling rates of qubits $Q_0$ and $Q_1$ to the feed line (microwave bath) are of the order of $\Gamma_{ex}/2\pi \approx 10$ Hz, showing the strong suppression of the radiation losses by the Purcell filter. In contrast, the experimentally measured relaxation rates are of the order of 1 kHz, confirming that the qubits are strongly under-coupled to the feed line ($\Gamma_{in}/2\pi \approx 1$ kHz, where $\Gamma_{in}$ is the qubit internal loss rate). Assuming the intrinsic qubit

bath temperature is negligible, we can estimate the microwave background photon noise ($n_{bg}$) required to have the residual excited-state probability of $P_E \approx 0.2$ by using

$$P_E = \frac{\Gamma_{ex} n_{bg}}{\Gamma_{ex}(2n_{bg} + 1) + \Gamma_{in}}. \qquad (2)$$

This results in $n_{bg} \approx 30$.

In the scenario of an abrupt thermal heating due to the pulse tube mechanical vibrations at the mixing chamber plate of the dilution refrigerator, the microwave bath would undergo a heating process analogous to that observed in the qubits. Consequently, the background microwave photon noise would be elevated to approximately $n_{bg} \approx 0.2$ during the transient period when the qubits reach the maximal excitation due to the pulse tube operation, as shown in Fig. 3d.

When it is characterized by a microwave detector with a quantum efficiency of $\eta$, the background microwave photon noise is measured to be $\eta n_{bg} + 1/2$ in the unit of quanta (see more details in "Methods" section). By using the relationship between the signal-to-noise ratio in single-shot qubit readout and the readout-induced qubit dephasing, we confirm that our microwave measurement achieves a quantum efficiency of $\eta > 0.2$ (see Supplementary Note 5). This is facilitated by the JTWPA, operated as a nearly quantum-limited phase-insensitive amplifier. If the microwave background noise dominantly heats up the qubits, the measured photon noise on top of the vacuum noise (1/2), therefore, needs to be increased to be of the order of $\eta n_{bg} \gtrsim 6$ and 0.04 for the two aforementioned scenarios, respectively. However, there is no increase observed at such a level in the background noise, as shown in Fig. 7d. This confirms that possible nonequilibrium dynamics of the microwave bath do not play a dominant role in the mechanically induced excitations of the qubits.

## Discussion

While we clearly observe that the transmon qubits suffer from correlated bit-flip errors due to the global nature of mechanical vibrations, the physical origin still remains uncertain. Here, we propose and discuss possible explanations for the mechanically induced excitations and quantum jumps of the qubits.

First, we can rule out the possibility that the qubits are excited by an abrupt thermal heating of the mixing chamber plate caused by a mechanical shock. In fact, this is not consistent with the non-thermal probability distribution of the qubit produced by the pulse tube mechanical vibrations (see Fig. 2f). Next, we verify that the mechanical vibrations do not affect the microwave environment around the qubit frequencies, although they could cause a triboelectric effect, generating low-frequency electrical noise in cables and degrading the coherence of a spin qubit[47]. Indeed, we record no fluctuation in the transmission coefficient and background noise of the feed line while the mechanical bursts excite the qubits (see Fig. 7). Furthermore, the absence of variation in the microwave background noise also supports the lack of heating of the mixing chamber plate (thermally coupled to the microwave environment via the attenuators). In addition, this is also consistent with the qubit excitations caused by a pure mechanical shock, as shown in Fig. 6, where possible electrical noise generated by the pulse tube cooler does not play an important role in the qubit excitations. Nevertheless, our characterization does not cover the frequency range for infrared photons, which are more sensitive to a small mechanical displacement in microwave components and can induce a qubit decay by breaking Cooper pairs[61].

We associate the mechanical sensitivity of the qubits with quasiparticle- and TLS-mediated interactions to their phononic baths. On one hand, high-energy phonons break Cooper pairs, resulting in nonequilibrium quasiparticles in the qubit electrodes[41,62]. The quasiparticles are quickly cooled down to the lowest energy level above the

superconducting energy gap, remaining there for several tens of milliseconds before recombining into Cooper pairs[29,34]. This nonequilibrium quasiparticle bath could be heated by scattering with the phonons produced by the pulse tube mechanical vibrations, resulting in excitations or relaxations of the qubits. On the other hand, TLSs couple to a qubit via their electric dipole-dipole interactions while coupling to a phononic bath via their strain potential[40]. Therefore, they can mediate between the qubit and the phononic bath, resulting in the mechanical sensitivity of the qubit. Finally, both a change in the strain[63] and the saturation of the TLS bath[64] can alter the coupling between the qubit and the TLSs, leading to a fluctuation in the qubit decay rate to the phononic bath. Additionally, the TLS model also explains the long-term fluctuations that are observed in the lifetimes of our qubits[26,27], as shown in Fig. 1g.

In summary, in this work, we presented a novel time-resolved measurement technique to study the mechanical sensitivity of long-lived transmon qubits, synchronized with the operation of the pulse tube cooler of a dilution refrigerator. Our results demonstrated that the mechanical vibrations generated by the pulse tube cooler induce dominant bit-flip errors in the qubits. Moreover, the global nature of the mechanical bursts on the multi-qubit device causes correlated errors among the qubits, which are detrimental to realizing large-scale quantum computing based on quantum error correction. While the origin of mechanical sensitivity of the qubits could not be established unequivocally, our observations are consistent with quasiparticle- and TLS-induced qubit decay models[40,41], and provide valuable insights into the loss mechanisms that limit the state-of-the-art qubit coherence.

Our findings suggest several strategies to mitigate the mechanical sensitivity of superconducting qubits, including the use of a suspended qubit substrate with phononic crystal structures[42–44] and sample packages that employ both mechanical isolation and thermal conductivity[45,46], in order to isolate superconducting qubits from mechanical vibrations. In addition, our results would emphasize the importance of vibration-free dilution refrigerators[47–49] in achieving long and stable coherence times in superconducting devices. Moreover, our measurement scheme revealed that superconducting qubits become out of equilibrium within a specific time window during the periodic pulse tube operation. This observation suggests a straightforward mitigation strategy: synchronizing qubit experiments with the pulse tube operation and acquiring data only when the pulse tube effect is minimal. We believe that these insights will be valuable for the development of next-generation superconducting-qubit technologies that pave the way for realizing fault-tolerant quantum computing.

## Methods
### System parameters
The system parameters of the multi-qubit device are summarized in Table 1.

The qubit frequencies ($\omega_q$), relaxation times ($T_1$), and dephasing times ($T_{2^*}$ and $T_{2e}$) are obtained as the averages of the long-term stability measurement data collected over 400 h, respectively, shown in Fig. S3 of SI, while the error bars are calculated as the standard deviations. The longest relaxation and dephasing times, shown in Fig. 1e, f, are observed in the first cooling down for the multi-qubit device, while the long-stability measurement is conducted in the following cooling down, where the relaxation and dephasing times are slightly degraded possibly due to additional oxidation of the Nb and Si surfaces. The anharmonicities ($\alpha$) are characterized by the E-F control in the time domain ($Q_0$ and $Q_1$) and the two-photon transition in the qubit excitation spectra ($Q_2$ and $Q_3$). We perform the F-state relaxation measurement for qubits $Q_0$ and $Q_1$ and obtain the F-state probability using the averaged readout complex amplitude for different delay times. The exponential fitting to the time trace resulted in the $T_1$ of the F state.

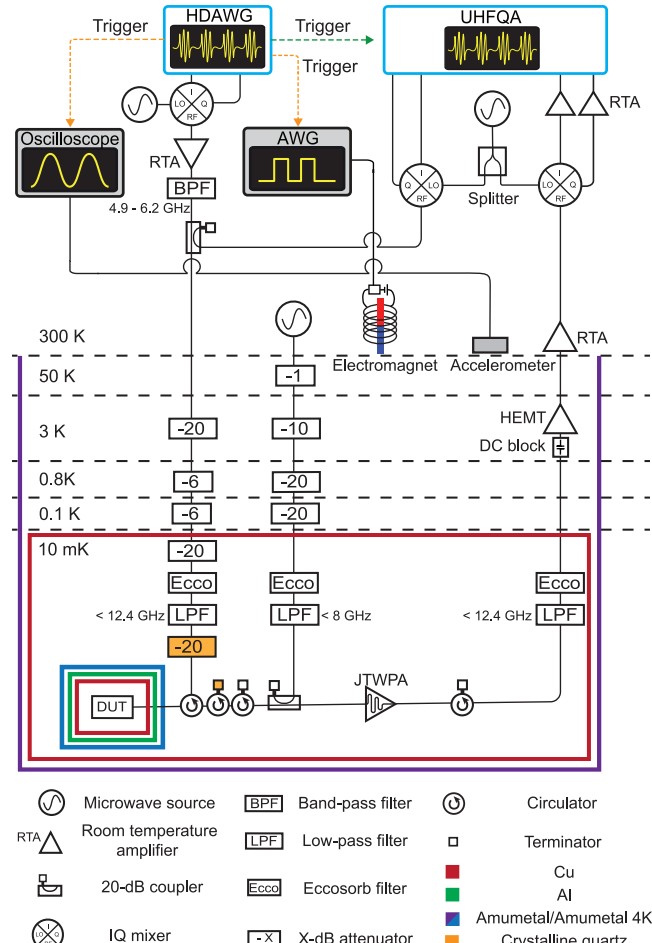

**Fig. 8 | Schematic of experimental setup.** Cryogenic wiring and room temperature measurement setup.

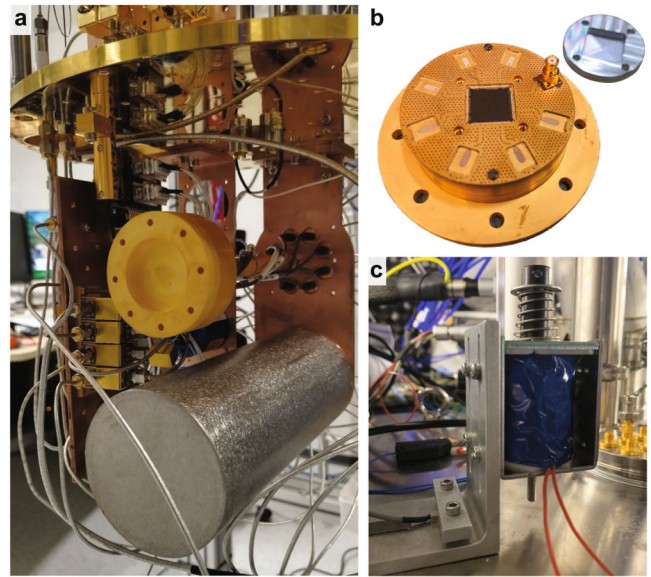

**Fig. 9 | Sample holder and package, electric hammer, and accelerometer. a** Two slots for sample holders. The DUT is mounted inside the multilayer shielding at the bottom slot. **b** Multi-qubit device wire-bonded to a PCB on a sample table. An Al lid is covered to suppress qubit radiation loss. **c** Electric hammer based on an electromagnet. An accelerometer is attached to the jig for the hammer.

The frequencies, external coupling rates, and internal loss rates of the readout resonators are characterized from the reflection spectra with the qubits in the $G$ states in continuous-wave (CW) measurement. Since the qubits are well cooled down to their ground states in our experimental setup, the effective intrinsic losses due to the qubit excitations are negligible[65]. The dispersive shifts for $Q_0$ and $Q_1$ are determined by the resonance frequency difference of the qubit-state-dependent reflection spectra of the resonators in the time-domain protocol (see Fig. S4a in SI), while those for $Q_2$ and $Q_3$ are obtained from the photon-number resolved qubit excitation spectra.

**Cryogenic wiring and room temperature measurement setup**
The experimental setup is shown in Fig. 8. The chip is fixed on a sample table made of gold-plated Cu, wire-bonded to a PCB based on coplanar waveguides sandwiched by double ground planes, and covered by an Al lid (see Fig. 9b). There is an air gap designed under the chip on the sample table to increase the frequency of spurious chip modes well above our working frequencies[66]. We thermally and mechanically anchor the device under test (DUT) to the mixing chamber stage (≈10 mK) of a dilution refrigerator (BlueFors BF-LD250). We isolate the DUT from environment fluctuations with multiple shielding: Al and Amumetal-4 K shields are used for reducing magnetic noise, while a Cu shield is used for thermalizing the qubit radiation field to the base temperature (see Fig. 9a). In addition to the standard equipment defined by BlueFors (300 K vacuum can and 50 K, 3 K, and 0.8 K radiation shields, not shown in Fig. 8), an outer Amumetal shield is equipped and thermalized to the vacuum can (300 K) while a Cu

radiation shield is thermalized to the 10 mK stage, covering the entire 10 mK setup. The input line to the DUT is equipped with a series of cryogenic attenuators (−72 dB in total) to suppress thermal noise from the higher temperature stages, while the output line is equipped with several isolators to prevent back heating from amplifiers. The readout signals are amplified by a Josephson traveling wave parametric amplifier (JTWPA) and a high electron mobility transistor (HEMT) amplifier in the dilution refrigerator, allowing us to realize a nearly quantum-limited microwave measurement. The continuous pump signal to operate the JTWPA is added to the readout chain after the DUT via a directional coupler. We optimize the pump power and frequency to maximize the signal-to-noise ratio around the readout frequencies, resulting in about a 20 dB gain. In addition, all the input, output, and pump lines are equipped with low-pass filters (LPFs) and eccosorb filters (Eccos) to reduce the contamination of high-frequency photons.

For multiplexed control and readout of the transmon qubits, we employ an arbitrary waveform generator (Zurich Instruments HDAWG) and a quantum analyzer (Zurich Instruments UHFQA) to generate, acquire, and analyze intermediate frequency (IF) pulse sequence. We up- and down-convert frequency-multiplexed IF signals using IQ mixers operated with continuous waves generated from a multi-channel microwave source (AnaPico APMS12G). The up-converted control signal is amplified, filtered to cut the output amplifier noise around the readout frequencies, and combined with the readout signal via a directional coupler. The frequency-multiplexed readout microwave signals are down-converted, amplified by room temperature amplifiers (RTA), digitized, and, demodulated by the UHFQA, leading to the $I$ and $Q$ quadratures for each readout frequency. The operation of the UHFQA, generating and digitizing the readout signals, is synchronized with the HDAWG via a trigger signal.

For monitoring the vibrations of the top plate of the dilution refrigerator, we use an oscilloscope (Keysight 1000X), operated synchronously with the HDAWG via a trigger signal. The acceleration of the top plate is continuously converted to a voltage signal by an accelerometer (KEMET VS-BV203-B) mounted on it (see Fig. 9c). The accelerometer can be activated with a 5 V DC voltage bias (not shown in Fig. 8). Upon a trigger signal from the HDAWG, the oscilloscope

starts to record the converted voltage signal. The recorded voltage signals are expressed in the unit of acceleration by using the sensitivity of 20 mV/m/s².

For artificially generating a pure mechanical shock on the top plate of the refrigerator, we use an electric hammer based on a circuit consisting of an electromagnet, a 15 V DC voltage bias, and an electrical switch (see Fig. 9c). When the electrical switch is on, a current flows in the electromagnet, accelerating the magnet core and resulting in a mechanical shock on the top plate. When the electrical switch is off, the magnet core is detached from the top plate by the elastic force of a spring. We use an AWG (Tektronix AFG3252C), operated synchronously with the HDAWG, to control the electrical switch by a pulsed voltage signal. In the experiment for Fig. 6, the electric hammer is activated periodically with 50 ms pulsed signals with a period of 1.5 s.

### Quantum efficiency of microwave measurement

To exclude possible responsibility of electrical noise or local heating for the qubit nonequilibrium dynamics, we study the effect of the pulse tube on the background noise of the microwave bath. To achieve nearly quantum-limited microwave measurement, we amplify microwave background noise interacting with the qubit device by the JTWPA and the HEMT amplifier in the dilution refrigerator, followed by another amplification, demodulation, and digital sampling at room temperature. The measurement outcome of an input noise ($n_{\mathrm{in}}$) can be simply described as

$$n_{\mathrm{out}} = C_{\mathrm{m}}(n_{\mathrm{in}} + n_{\mathrm{m}}), \qquad (3)$$

where $C_{\mathrm{m}}$ and $n_{\mathrm{m}}$ are the total scaling factor and the effective input-referred noise, respectively, which are uniquely determined by the combined contributions of propagation losses, amplifier gains, and amplifier-added noises in the measurement chain. Note that the expression is described in the unit of quanta.

For convenience, the outcome $n_{\mathrm{out}}$ is normalized by the output noise for the vacuum input ($n_{\mathrm{in}} = 1/2$) and re-scaled to 1/2, resulting in

$$\bar{n}_{\mathrm{out}} = \eta n_{\mathrm{in}} + (1 - \eta)\frac{1}{2}, \qquad (4)$$

where a quantum efficiency is defined as

$$\eta = \frac{1/2}{1/2 + n_{\mathrm{m}}}. \qquad (5)$$

Thus, the full microwave measurement chain can be effectively considered as a single quadrature detector with a quantum efficiency of $\eta$, or an insertion of a beam splitter with a transmittance $\eta$ in front of an ideal quadrature detector[67]. Since our measurement chain adopts a phase-insensitive amplification enabled by the JTWPA as a pre-amplifier, the input-referred noise is limited by $n_{\mathrm{m}} \geq 1/2$, resulting in $\eta \leq 1/2$. In our analysis, we normalized the outcome by the non-time-resolved (averaged) noise, corresponding to the possible lowest noise since there is no significant increase in the time-resolved noise data. This is valid as long as the measurement bath is close to being in the vacuum at least when the pulse tube effect is minimal. A finite thermal photon noise would affect the estimated quantum efficiency[68], which we can safely assume is quite small given the heavily attenuated, filtered, and isolated input and output lines.

Within this formalism, the output for the input of thermal background noise ($n_{\mathrm{in}} = 1/2 + n_{\mathrm{bg}}$) is described as

$$\bar{n}_{\mathrm{out}} = \eta n_{\mathrm{bg}} + \frac{1}{2}, \qquad (6)$$

which are shown in Fig. 7d in the main text.

We calibrate the attenuation in the dilution refrigerator based on photon-number resolved qubit spectra. With this, we estimate the readout photon flux incoming to the readout cavity, resulting in the readout-induced qubit dephasing rate. By taking the ratio of the signal-to-noise ratio rate of the single-shot readout to the estimated dephasing rate, we characterize the quantum efficiency of our measurement chain as $\eta > 0.2$ (see more details in Supplementary Note 5).

## Data availability
The data used to produce the plots within this paper are available on Zenodo (https://doi.org/10.5281/zenodo.11034817). All other data are available from the corresponding authors upon request.

## Code availability
The code used to produce the plots within this paper will be available on Zenodo (https://doi.org/10.5281/zenodo.11034817).

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

## Acknowledgements

We thank Sebastian Cozma and Pasquale Scarlino for helping with designing the sample package and Yang Xu for developing the Nb deposition and etching, respectively. Moreover, we thank MIT Lincoln Laboratory and William D. Oliver for providing the JTWPA. This work was supported by the European Research Council (ERC) grant No. 835329

(ExCOM-cCEO), as well as the Swiss National Science Foundation (SNSF) under grant No. 204927 and the NCCR QSIT grant No. 51NF40-185902. S.K. acknowledges support from the EU H2020 research and innovation program under the Marie Sklodowska-Curie grant agreement No. 101033361 (QuPhon). M.S. acknowledges support from the EPFL Center for Quantum Science and Engineering postdoctoral fellowship. All devices were fabricated in the Center of MicroNanoTechnology (CMi) at EPFL.

## Author contributions

S.K. conceived the experiment. S.K. and M.C. designed the device and developed the fabrication process with the assistance of X.W. S.K. and J.P. established the measurement setup and performed the measurements. S.K. and J.P. analyzed the data with the assistance of M.C. S.K., J.P., M.C., M.S., and T.J.K. wrote the manuscript with feedback from all the coauthors. T.J.K. and S.K. supervised the project.

## Competing interests

The authors declare no competing interests.
