## [Peer Review File · Nature Communications]

REVIEWER COMMENTS

Reviewer #1 (Remarks to the Author):

I have read with great interest the manuscript by Kono, Pan, Chegnizadeh et al., in which they report convincing evidence that vibrations induced by commonly used pulse-tube cryostats impact the performance of superconducting quantum bits. Their findings are directly relevant for the community of superconducting quantum bits, but not only. Similarly to the recent revelations about the role of ionizing radiation in microelectronics quantum devices, the current work will impact many solid-state quantum devices beyond the superconducting qubit community. While these results do not come as a shocking surprise for the experts in the field, it is valuable and impactful to see such clear evidence supported by state-of-the-art experiments.

The text is well written and easy to read, supported by clear and meaningful figures. I am convinced Nature Communications is the appropriate platform to present these results, and I anticipate that I will recommend publication once the authors will have addressed the technical points below, some of which are quite important to fix:

1) I recommend shortening the title by removing "with relaxation times exceeding 0.4 milliseconds", for two reasons: A) as clearly visible in Fig. 1g the measured relaxation times are usually between 50 and 200 μ s, with occasional excursions outside this range (both above and below), B) I find the shorter version of the title stronger and more appealing to a general audience.

2) Related to point #1 above, in the abstract, the authors state "Here, we realize ultra-coherent superconducting qubits with lifetimes exceeding 0.4 ms and ...". This is not an accurate representation of the measured data presented in the text. A more appropriate wording would be "with relaxation times between 50 and 200 μ s, with fluctuations outside this range (both above and below)."

3) In the Results section, the authors title a sub-section "Ultra-coherent transmon qubits". I feel a duty to push back against this inflation of superlatives, which is disappointing and ridiculous in the same time. If this is "ultra-coherent", how are we going to call the qubits which actually pass the minimum bar for error correction? I advise the authors to tone-down this formulation in the title and in the section text.

4) The error probabilities for the G and E states are stated to be 0.1% and 3% respectively. Looking at the evidence provided in the supplementary information, these numbers are wrong. For the best qubit, i.e. Q0, from Fig. S5 I get 5% readout error for E, while for Q1 I get 7-8% error for E. For G the numbers are about 10x off (i.e. 1% error instead of 0.1%). The other qubits, with lower T1, likely have even lower fidelity, but no information is given in S5.

To make my point clear, I am attaching FigS5.jpg, in which the authors can directly see what I mean. For example, for qubit Q0, for E I estimate an error of 100 counts in 2000 (i.e. 1 in 20, or 5%) and for G I estimate an error of 24 counts in 2040 (i.e. 1.2%).

5) An additional small remark about figure S5: it would be useful to align the X axis between the scatter plots (on the top row) and the histograms (on the bottom row), to make the X-axis projection easy to follow by eye.

6) Please note that in the case of Q1, the I projections for states E and F overlap, which can introduce various artifacts. This could be one reason why the Gaussian fit is off for Q1 in the excited state.

7) In the attached I_histograms.jpg I point to several irregularities in the double Gaussian fit, which I can not explain. It looks like some points of the model line moved arbitrarily in the plot.

8) I think it is relevant to mention in the main text what is the photon number occupancy in the readout resonator during the readout pulse? I could not find this information even in the supplementary.

9) In Fig. 4, I do not understand where the 2π division factors come from in the formulas for the qubit transition rates (this is not the case in Ref. [52] which the authors cite for their methodology). The resulting numbers for the decay rates are confusing, because they suggest a longer T1 compared to a T1 value extracted from a standard exponential fit.

10) I can extract a T1 time from the dwell time histograms presented in Fig 4 (for example see T1_from_dwell_time.jpg for Fig. 4e – excited state) and I get a T1 value for Q0 between 70 and 100 microseconds. Why are these T1s not consistent with the free decay presented in Fig. 1? The authors should comment on this in the main text.

Reviewer #2 (Remarks to the Author):

The authors carried out a novel experiment where they compared qubit measurements with readings from an accelerometer mounted on the top plate of a dilution refrigerator. From the time correlation between the two they found non-equilibrium qubit populations correlated to the period of the pulse tube of the dilution refrigerator. The method is novel, and the quality of the data and the analysis is high. The conclusion that the pulse tube may affect the performance of superconducting quantum processors is highly relevant in the technological sense. In addition, their qubits have amazing coherence properties for a device using Nb electrodes on Si. There is clearly a correlation between the pulse tube and the qubit dynamics, however, I do not believe that the data presented is enough to support the claim that “*mechanical* shocks from the pulse tube induce nonequilibrium dynamics”. This article may deserve publication in Nature Communications if the authors could address several major reservations listed below:

1. Fig 3d shows that the non-equilibrium populations appear primarily between 0.15 s and 0.35 s during each pulse tube cycle, while in figure 3c there are also peaks in acceleration at ~ 0.1 s and 0.4 s. If the non-equilibrium dynamics is truly caused by mechanical means, then the population of the qubit should follow a similar pattern as the accelerometer measurements (as seen in the electric hammer experiment).
2. The dominant peak in fig 2c is 2×1.4 Hz. This suggests that both gas in and gas out generate mechanical shock. In comparison, the dominant peak in fig 2e is 1×1.4 Hz. This does agree with fig 3d, but it does not support that mechanical shocks is the main cause here.
3. The authors mentioned that the accelerometer is mounted at “the top plate” of the dilution refrigerator. Do you mean it is mounted at the very top of the fridge? Perhaps what matters most is the vibrations at the mixing chamber level. Does the acceleration look the same at the mixing plate? Would “gas in” and “gas out” both generate vibrations at the mixing plate of a similar magnitude? It’s possible that maybe only “gas in” can generate vibrations at the mixing plate while both “gas in” and “gas out” generate vibrations at the top of the fridge. This is relatively easy to test even at close to room temperature.
4. Fig 6 shows a clear correlation between the acceleration caused by the hammer and the excited state of the population. This figure does support that it is possible for mechanical noise from the pulse tube to influence the qubit population. However, I do not believe that this supports the claim that “any possible electrical noise generated from the pulse tube cooler is not dominantly involved in the qubit excitations”.
5. In addition, the acceleration generated by the hammer is up to ~ 200 m/s², which is much larger compared to up to ~ 10 m/s² as from the pulse tube as shown in for example fig 3c. But the excited

state population due to the pulse tube can reach 10^{-1} according to fig 3d while a much larger shock from the hammer can only generate a population of $\sim 10^{-2}$ from fig 6c. Therefore, the figure for the electric hammer experiment shows that the out-of-equilibrium population of the qubit cannot be explained by mechanical shocks alone.

Here're also some minor comments:

1. In figure 2d and 3d, the authors showed that the qubit is out of equilibrium during a ~ 0.15 s long interval every ~ 0.7 s (1.4 Hz frequency). My question is: if you exclude the data during these ~ 0.15 s out-of-equilibrium intervals, would you be able to distinguish between when the pulse tube is on vs. when it is off? This might work as a simple technique to improve qubit performance.

2. Since the study involves F states, it would be more appropriate to include at least the T1 value of the F states in table 1.

3. The second sentence under "Mechanically induced correlated excitations" reads a little weird to me. "...readout quadrature outcomes of approximately 3×10^5 and 10^7 when the pulse tube cooler is off and on...". I recommend changing it to "... 3×10^5 and 10^7 measurements..." or something similar.

4. In the second paragraph under "Mechanically induced correlated excitations", the authors used a notation " 2^{12} ". I recommend simply using 4096 (this is not a big number) or just ~ 4100 unless the 12th exponent carried some special meaning. Similarly, " 2^{19} " is used under "Mechanically induced correlated quantum jumps".

5. Can the authors elaborate on their claim that larger non-time-resolved MI implies "the time-resolved events are not strongly correlated, but macroscopic parameters, such as the intrinsic decay rates or bath occupations, are changing synchronously for both the qubits, causing the correlated errors in the non-time resolved analysis"?

6. Time interval in fig 5c is defined by the authors as "When the qubits are continuously monitored for a certain time interval, an error, corresponding to a state transition, will occur with a finite probability". Does this mean that a new sequence is developed which results in fig 5c, or did you dice the data using the sequence shown in fig 4a into chunks with length equal to the time interval to produce fig 5c? Also, is the MI shown in 5c an average value during the whole 0.7 period of the pulse tube cycle?

7. I found that notations of " 10^{-n} " and " e^{-n} " are used interchangeably in the manuscript. I recommend sticking to one notation throughout the text.

8. The authors claim that "We associate the mechanical sensitivity of the qubits with TLS- and quasiparticle-mediated interactions to their phononic baths" based on the analysis of the fluctuation of T1 overtime when the pulse tube is on. The fluctuation analysis can only prove that the qubit is TLS-limited similar to most highly coherent transmon qubits. From this analysis alone and statement such as "high-energy phonons break Cooper pairs", I don't think one can unambiguously associate the observation to TLS or quasiparticle.

Reviewer #3 (Remarks to the Author):

Kono et al. fabricate superconducting qubits with Nb electrodes that have relaxation times of several 100 microseconds. They perform time-resolved measurements of the population of the ground, first and second excited states of these qubits. They additionally characterise the vibrations of their dilution refrigerator caused by the pulse tube cooling unit by means of an accelerometer attached to the 300K plate of the instrument. The authors find a positive correlation between high vibrational noise and high excited state population as well as increased state switching rates. They perform control experiments to exclude electric and triboelectric noise (the latter only at a single frequency) and state that their observations would be compatible with TLS and quasiparticle-mediated qubit decay to a phononic bath.

I find the manuscript well-written, technically sound, and original. The data is thoroughly analysed and relevant control experiments are included. That being said, before recommending publication, I would like to see the following comments and questions addressed:

1) Given that the authors state in the abstract that "Our observations are consistent with qubit dynamics induced by two-level systems and quasiparticles, thereby deepening our understanding of the qubit error mechanisms.", this point should be argued more conclusively in the manuscript. While the experiment is indeed original and interesting, there is no direct evidence indicating that TLSs and quasiparticles link the mechanical vibration and qubit dynamics. This statement is based on a rather speculative paragraph at the end of the discussion. I think this hypothesis can certainly be mentioned, but should not be used as a central claim in the abstract. Therefore, the statement could be removed in an effort to not be misleading while, in my opinion, not detracting from the interest of the work.

2) Y-axis of Fig.1 e and f as well as text: "excitation probability" could be read as "probability of exciting the qubit" as opposed to "probability of finding the qubit in the excited state" or in short: "excited-state probability". I believe the latter is what is meant here.

3) Fig2c: Could the authors comment on what the highest frequency harmonics extracted from the accelerometer data are?

4) Fig2f,d and all other relevant plots: I suggest to normalise the quadrature axis to the standard deviation of the Gaussian fit corresponding to the qubit ground state. This adds more information compared to arbitrary units.

- 5) "readout quadrature outcomes of approximately $3 \times 10^{**5}$ and 10^{**7} when...": missing word: "measurements". Please justify why fewer measurements were taken in the "off" state than in the "on" state.
- 6) Fig3a: The number of measurements is given in the text, but given the normalisation of the colour scale that information should also be in the figure caption.
- 7) Fig3 caption: typo: "(red shaded)" is given twice.
- 8) "accumulate a sufficient number of the single-shot outcomes at an arbitrary time of interest within the vibrational period to obtain the time-resolved residual excitation probabilities of the qubits." Please comment on what sets that sufficient number and add a one-standard deviation confidence interval on the time-resolved mutual information plotted in Figure 3e.
- 9) "The time-resolved MI (blue solid line) is always smaller than the non time- resolved value (orange dashed line), implying the time-resolved events are not strongly correlated, but macroscopic parameters, such as the intrinsic decay rates or bath occupations, are changing synchronously for both the qubits, causing the correlated errors in the non-time resolved analysis." Could the authors elaborate on this further? How does this difference in time-resolved and non-time-resolved MI change with the timespan chosen for the time bin of the non-time-resolved MI?
- 10) Section: "Mechanically-induced correlated jumps": In this section is the F-state population added to the E-state population or neglected? Please comment on how this choice influences the results in particular the error matrix Fig5d.
- 11) Fig5 caption: "The shaded regions in (a) and (b) depict the fitting errors." What exactly is meant by fitting errors here? Could you formulate this in terms of residual standard deviation?
- 12) "we can characterise the microwave transmission ratio and background noise at a time of interest within the period of the vibrational noise". Please elaborate further on how exactly the background noise is extracted. In particular, how is the y-axis of Fig7b calibrated in units of noise quanta?
- 13) Fig7b: this figure is very busy and pixelated when zooming in. Please provide the data in higher resolution.
- 14) Fig8: Is there indeed a Cryoperm shield thermalised to the 50K plate and a Cu shield surrounding the entire 10mK stage setup? If so, please comment on the reasons for this. What about shielding thermalised to 3K and 0.8K?

Reply to Referee Comments

Manuscript “Mechanically induced correlated errors on superconducting qubits with relaxation times exceeding 0.4 milliseconds”

Below we repeat the Referees’ reports verbatim, with our reply in *blue* and actions taken in the manuscript in *red bullets*.

Referee #1:

I have read with great interest the manuscript by Kono, Pan, Chegnizadeh et al., in which they report convincing evidence that vibrations induced by commonly used pulse-tube cryostats impact the performance of superconducting quantum bits. Their findings are directly relevant for the community of superconducting quantum bits, but not only. Similarly to the recent revelations about the role of ionizing radiation in microelectronics quantum devices, the current work will impact many solid-state quantum devices beyond the superconducting qubit community. While these results do not come as a shocking surprise for the experts in the field, it is valuable and impactful to see such clear evidence supported by state-of-the-art experiments.

The text is well written and easy to read, supported by clear and meaningful figures. I am convinced Nature Communications is the appropriate platform to present these results, and I anticipate that I will recommend publication once the authors will have addressed the technical points below, some of which are quite important to fix:

Reply:

We thank Referee #1 very much for their interest in our work and the positive evaluation. We deeply appreciate their careful reading and comments. Here, we addressed the comments and modified the manuscript accordingly.

1) I recommend shortening the title by removing "with relaxation times exceeding 0.4 milliseconds", for two reasons: A) as clearly visible in Fig. 1g the measured relaxation times are usually between 50 and 200 us, with occasional excursions outside this range (both above and below), B) I find the shorter version of the title stronger and more appealing to a general audience.

Reply:

We thank Referee #1 for their suggestion. We observed prominent qubit mechanical sensitivity only with two of the best long-lived superconducting qubits, implying that the exceptionally long lifetimes are also an essential part of our work. Therefore, we would like to keep "with relaxation times exceeding 0.4 milliseconds" in the title to summarize our work very well and to attract readers who are interested in the long relaxation time of superconducting qubits.

2) Related to point #1 above, in the abstract, the authors state “Here, we realize ultra-coherent superconducting qubits with lifetimes exceeding 0.4 ms and ...”. This is not an accurate representation of the measured data presented in the text. A more appropriate wording would be "with relaxation times between 50 and 200 us, with fluctuations outside this range (both above and below)."

Reply:

We agree with their suggestion. We modified the abstract to be more precise.

Modification:

✓ We mentioned both the longest and average lifetimes in the abstract.

3) In the Results section, the authors title a sub-section "Ultra-coherent transmon qubits". I feel a duty to push back against this inflation of superlatives, which is disappointing and ridiculous in the same time. If this is "ultra-coherent", how are we going to call the qubits which actually pass the minimum bar for error correction? I advise the authors to tone-down this formulation in the title and in the section text.

Reply:

We agree with their suggestion.

Modification:

- ✓ We replace “ultra-coherent” with “long-lived.”

4) The error probabilities for the G and E states are stated to be 0.1% and 3% respectively. Looking at the evidence provided in the supplementary information, these numbers are wrong. For the best qubit, i.e. Q0, from Fig. S5 I get 5% readout error for E, while for Q1 I get 7-8% error for E. For G the numbers are about 10x off (i.e. 1% error instead of 0.1%). The other qubits, with lower T1, likely have even lower fidelity, but no information is given in S5.

To make my point clear, I am attaching FigS5.jpg, in which the authors can directly see what I mean. For example, for qubit Q0, for E I estimate an error of 100 counts in 2000 (i.e. 1 in 20, or 5%) and for G I estimate an error of 24 counts in 2040 (i.e. 1.2%).

Reply:

The total error of the single-shot qubit readout outcomes presented in Fig. S5b (original) can be divided into three contributions: the state-preparation error, the separation error, and the state-flip error. The readout error defined in the manuscript includes only the separation error and the state-flip error. As explained in Supplementary Note 3, we characterized the separation error using the 2D Gaussian distributions of the single-shot readout quadrature amplitudes for the *G*, *E* and *F* states, while we characterized the state-flip error using the state-transition rates during continuous monitoring. We assume that the state-preparation errors for *G* and *E* (*F*) states can be excluded from the readout errors, which are dominated by the residual thermal excitation and the control errors, respectively.

Modification:

- ✓ We added more detailed explanations to Supplementary Note 3.
- ✓ Note that we excluded Fig. S5b (original) from the revised manuscript since we use the thresholds to distinguish between *G*, *E*, and *F* in the complex plane.

5) An additional small remark about figure S5: it would be useful to align the X axis between the scatter plots (on the top row) and the histograms (on the bottom row), to make the X-axis projection easy to follow by eye.

Modification:

- ✓ We excluded Fig. S5b (original) because we no longer rely solely on the I quadrature.

6) Please note that in the case of Q1, the I projections for states E and F overlap, which can introduce various artifacts. This could be one reason why the Gaussian fit is off for Q1 in the excited state.

7) In the attached I_histograms.jpg I point to several irregularities in the double Gaussian fit, which I can not explain. It looks like some points of the model line moved arbitrarily in the plot.

Reply:

Thank Referee #1 for their careful check on our plots. We realize that there was a bug in the generation process of the figures (not in our analysis). Indeed, the fitting curves are strangely deformed, showing inaccurate curves. We really apologize for the confusion. This bug simply explains the discrepancy between the experimental data and fitting.

Modification:

- ✓ We excluded Fig. S5b (original) because of the reason explained above. Nevertheless, the same analysis is performed to obtain the occupation probabilities of the *G*, *E*, and *F* states for Q_0 in Fig. 2f, showing no fitting discrepancy.

8) I think it is relevant to mention in the main text what is the photon number occupancy in the readout resonator during the readout pulse? I could not find this information even in the supplementary.

Reply:

We calibrated the incoming readout photon flux based on the photon number resolved qubit spectrum, resulting in the estimation of the average photon number in the readout cavity with the qubit in the *G*, *E* and *F* states during the readout, respectively (see more details in Supplementary Note 5).

Modification:

- ✓ We added the average photon numbers in the readout cavity for the G , E , and F states to Table 1 in the method section of the main text.

9) In Fig. 4, I do not understand where the 2π division factors come from in the formulas for the qubit transition rates (this is not the case in Ref. [52] which the authors cite for their methodology). The resulting numbers for the decay rates are confusing, because they suggest a longer T_1 compared to a T_1 value extracted from a standard exponential fit.

Reply:

We believe that we are following the standard procedure (including the reference the referee mentioned) to extract the decay time. We fit the histogram to the exponential curve defined as $A \exp(-\Gamma t) + B$, where Γ is a transition rate, and A and B are additional fitting parameters. The results are shown in the unit of Hz (i.e., $\Gamma/2\pi$). In this case, the decay time is calculated as $T = 1/\Gamma$.

10) I can extract a T_1 time from the dwell time histograms presented in Fig 4 (for example see $T1_from_dwell_time.jpg$ for Fig. 4e – excited state) and I get a T_1 value for Q0 between 70 and 100 microseconds. Why are these T_1 s not consistent with the free decay presented in Fig. 1? The authors should comment on this in the main text.

Reply:

As shown in Fig. S5c (revised), our fitting results in $\Gamma_E/2\pi = 1.6$ kHz, approximately corresponding to $T_1 \sim 1/\Gamma_E \sim 100 \mu s$. Therefore, the referee's estimation is consistent with ours. The reason why the T_1 during continuous monitoring is shorter than that of the free evolution is due to the finite back-action of the readout, i.e., the readout-induced state flip of the qubit.

Modification:

- ✓ We added comments on page 15 of the marked-up main text.

Referee #2:

The authors carried out a novel experiment where they compared qubit measurements with readings from an accelerometer mounted on the top plate of a dilution refrigerator. From the time correlation between the two they found non-equilibrium qubit populations correlated to the period of the pulse tube of the dilution refrigerator. The method is novel, and the quality of the data and the analysis is high. The conclusion that the pulse tube may affect the performance of superconducting quantum processors is highly relevant in the technological sense. In addition, their qubits have amazing coherence properties for a device using Nb electrodes on Si. There is clearly a correlation between the pulse tube and the qubit dynamics, however, I do not believe that the data presented is enough to support the claim that “*mechanical* shocks from the pulse tube induce nonequilibrium dynamics”. This article may deserve publication in Nature Communications if the authors could address several major reservations listed below:

Reply:

We appreciate the positive assessments of the novelty and importance of our work. We agree that it is really challenging to directly identify the microscopic origin of the mechanical sensitivity of superconducting qubits. Nevertheless, we believe that we could exclude possible other electrical mechanisms caused by a side effect of mechanical shocks, e.g., triboelectric noise and local heating using well-controlled measurements, including the non-thermal distribution of the qubit and the unaffected microwave bath, etc. These observations result in the conclusion that nonequilibrium qubit dynamics can be induced by mechanical shocks.

1. Fig 3d shows that the non-equilibrium populations appear primarily between 0.15 s and 0.35 s during each pulse tube cycle, while in figure 3c there are also peaks in acceleration at ~ 0.1 s and 0.4 s. If the non-equilibrium dynamics is truly caused by mechanical means, then the population of the qubit should follow a similar pattern as the accelerometer measurements (as seen in the electric hammer experiment).

Reply:

We think that it could be possible that the time-resolved excited-state probability does not exactly follow the time trace of the acceleration of the vibrational noise presented in our manuscript. This is because the measurement frequency range of the accelerometer is up to 15 kHz from the specification (acceleration signals can be experimentally detected up to 50 kHz to a certain extent) while superconducting qubits need to be excited by GHz- or higher frequency vibrational noise as long as a multi-phonon excitation process and local heating do not play a dominant role. We believe that such a higher-order multi-phonon process bridging between kHz and GHz is negligible. In addition, we exclude the possibility of local heating by characterizing microwave background noise in a time-resolved fashion (see Fig. 7). Based on these observations, we believe that higher frequency vibrational noise that cannot be detected with the accelerometer could be responsible for the qubit nonequilibrium dynamics. Nevertheless, the low-frequency pulse tube vibrational noise, as illustrated in the main text, is useful for understanding the characteristics of the pulse tube operation. Specifically, it aids in time-aligning the outcomes of single-shot qubit readouts with respect to the pulse tube operation.

Modification:

✓ We added a more detailed explanation on pages 11 of the marked-up main text.

2. The dominant peak in fig 2c is 2×1.4 Hz. This suggests that both gas in and gas out generate mechanical shock. In comparison, the dominant peak in fig 2e is 1×1.4 Hz. This does agree with fig 3d, but it does not support that mechanical shocks are the main cause here.

Reply:

Indeed, this also surprised us. Upon this question, we carefully analyzed the vibrational signals generated by the pulse tube using a time-frequency analysis, revealing there is a difference between the gas-flow-in and out phases. This could be a piece of indirect evidence that the two different operation phases affect the qubit dynamics differently. However, a clear correlation between the time-resolved excited-state probabilities and the time-frequency analysis on the pulse tube vibrational noise is not observed up to 50 kHz. This could be because superconducting qubits can be excited by gigahertz or higher frequency vibrational noise that cannot be measured by the accelerometer. Although a further elaborate measurement may be required to uncover the microscopic origin more rigidly, we believe that the current manuscript is sufficiently convincing by showing the effect of the pulse tube on superconducting qubits for the first time.

Modification:

✓ We added a more detailed explanation on pages 11 of the marked-up main text.

✓ We added Fig. 3c to the main text.

3. The authors mentioned that the accelerometer is mounted at “the top plate” of the dilution refrigerator. Do you mean it is mounted at the very top of the fridge? Perhaps what matters most is the vibrations at the mixing chamber level. Does the acceleration look the same at the mixing plate? Would “gas in” and “gas out” both generate vibrations at the mixing plate of a similar magnitude? It’s possible that maybe only “gas in” can generate vibrations at the mixing plate while both “gas in” and “gas out” generate vibrations at the top of the fridge. This is relatively easy to test even at close to room temperature.

Reply:

We really appreciate the fruitful suggestion, which provides insights into our interpretation of the results. Indeed, the presented acceleration data were taken at the very top of the fridge. As the referee suggested, we characterized the vibrational noise at the qubit device when the dilution refrigerator was warmed up and open. As shown in Fig. S12a, only one of the two operation phases of the pulse tube shows a significant peak in the time trace of the vibrational noise, providing the indirect reason why only the gas-flow phase can contribute to the qubit nonequilibrium dynamics.

Modification:

✓ We added Supplementary Note 7, including the detailed explanations and Fig. S12.

4. Fig 6 shows a clear correlation between the acceleration caused by the hammer and the excited

state of the population. This figure does support that it is possible for mechanical noise from the pulse tube to influence the qubit population. However, I do not believe that this supports the claim that “any possible electrical noise generated from the pulse tube cooler is not dominantly involved in the qubit excitations”.

Reply:

We agree that it is challenging to strictly confirm that possible electrical noise from the pulse tube does not affect the nonequilibrium qubit dynamics. Nevertheless, we observed the transmission coefficient and background noise of the qubit control feedline do not show any significant variation upon the pulse tube mechanical shocks (see Fig. 7). Moreover, we observed a pure mechanical shock generated by an electric hammer can excite the qubits (see Fig. 6). These two observations strongly support the statement that mechanical shocks directly involve the nonequilibrium qubit dynamics. Upon the referee’s comments regarding the hammer experiment, we raised one possible concern that the electric hammer operation would locally heat the qubit environment or generate electrical noise that may excite the qubits. To address the concern, we took additional measurements as follows. First, we observed that the microwave transmission coefficient and background noise are not influenced by controlled mechanical shocks generated by the electric hammer, implying that local heating and electrical noise do not involve the qubit nonequilibrium dynamics (see Figs. S10c and d). Second, we performed the electric hammer experiment while the hammer was detached from the top of the fridge, where only possible electrical noise from the hammer operation could play a role. As a result, we confirmed that the qubits are not excited in this configuration (see Fig. S10b). These two additional results further convincingly support the statement that possible electrical noise from control electronics for the fridge operation, the microwave measurement, and the electric hammer operation does not involve the nonequilibrium qubit dynamics.

Modification:

✓ We added Supplementary Note 6, including the detailed explanations and Fig. S10.

5. In addition, the acceleration generated by the hammer is up to $\sim 200 \text{ m/s}^2$, which is much larger compared to up to $\sim 10 \text{ m/s}^2$ as from the pulse tube as shown in for example fig 3c. But the excited state population due to the pulse tube can reach 10^{-1} according to fig 3d while a much larger shock from the hammer can only generate a population of $\sim 10^{-2}$ from fig 6c. Therefore, the figure for the electric hammer experiment shows that the out-of-equilibrium population of the qubit cannot be explained by mechanical shocks alone.

Reply:

We thank Referee #2 for raising the good point. One of the reasons why the acceleration generated by the hammer is much larger than one from the pulse tube is that the accelerometer is attached next to the hammer on the top of the fridge (see Fig. 9) while the heads of the pulse tube are located at the 50-K and 4-K stages. This would significantly change the sensitivity of the vibration measurement for each vibration source. Upon this comment, we directly compare the accelerations from the hammer and the pulse at the qubit sample holder at room temperature. As shown in Fig. S12, the strength of the pulse tube vibrational noise is comparable to or slightly less than that of the electric hammer, which is still not consistent with the significantly larger qubit excitation induced by the pulse tube cooler. However, we believe that superconducting qubits could be excited by GHz- or higher frequency vibrational noise, which is challenging to measure with a conventional accelerometer. Nevertheless, we believe that on-chip correlation measurement of superconducting qubit dynamics and high-frequency vibrational noise will be a very interesting future work.

Modification:

✓ We added Supplementary Note 7, including the detailed explanations and Fig. S12.

Here’re also some minor comments:

1. In figure 2d and 3d, the authors showed that the qubit is out of equilibrium during a $\sim 0.15 \text{ s}$ long interval every $\sim 0.7 \text{ s}$ (1.4 Hz frequency). My question is: if you exclude the data during these $\sim 0.15 \text{ s}$ out-of-equilibrium intervals, would you be able to distinguish between when the pulse tube is on vs. when it is off? This might work as a simple technique to improve qubit performance.

Reply:

We agree with the idea from Referee #2. Indeed, the excited-state probability when the pulse tube effect is minimal is comparable with the one with the pulse tube off. This naturally suggests a novel mitigation strategy to synchronize qubit experiments with the pulse tube operation to obtain data only when the pulse tube effect is minimal.

Modification:

✓ We added the suggested mitigation strategy to the outlook section of the main text.

2. Since the study involves F states, it would be more appropriate to include at least the T1 value of the F states in table 1.

Reply:

We ran the *F*-state relaxation measurement for qubits Q_0 and Q_1 and obtained the *F*-state probability using the averaged readout complex amplitude for different delay times. The exponential fitting to the time trace resulted in the T_1 of the *F* state (~ 0.1 ms).

Modification:

✓ We included the T_1 of the *F* state in Table 1 in the method section of the main text.

3. The second sentence under “Mechanically induced correlated excitations” reads a little weird to me. “...readout quadrature outcomes of approximately 3×10^5 and 10^7 when the pulse tube cooler is off and on...”. I recommend changing it to “... 3×10^5 and 10^7 measurements...” or something similar.

Modification:

✓ We corrected them accordingly.

4. In the second paragraph under “Mechanically induced correlated excitations”, the authors used a notation “ 2^{12} ”. I recommend simply using 4096 (this is not a big number) or just ~ 4100 unless the 12th exponent carried some special meaning. Similarly, “ 2^{19} ” is used under “Mechanically induced correlated quantum jumps”.

Modification:

✓ We corrected them accordingly.

5. Can the authors elaborate on their claim that larger non-time-resolved MI implies “the time-resolved events are not strongly correlated, but macroscopic parameters, such as the intrinsic decay rates or bath occupations, are changing synchronously for both the qubits, causing the correlated errors in the non-time resolved analysis”?

We suggest that the pulse tube shock causes the correlated time variations of the qubit transition rates (i.e., the decay rates and the bath occupations), resulting in a correlation between the qubit excitations when the data are analyzed in a non-time-resolved manner. Namely, both the qubits are more frequently excited at one time than at another time during the period of the pulse tube operation. However, this does not mean that individual excitation events are strongly correlated between the two qubits (, like a two-mode squeezing), supported by the fact that the time-resolved mutual information is always below the one analyzed in the non-time-resolved manner.

Modification:

✓ We modified the explanation to be clear.

6. Time interval in fig 5c is defined by the authors as “When the qubits are continuously monitored for a certain time interval, an error, corresponding to a state transition, will occur with a finite probability”. Does this mean that a new sequence is developed which results in fig 5c, or did you dice the data using the sequence shown in fig 4a into chunks with length equal to the time interval to produce fig 5c? Also, is the MI shown in 5c an average value during the whole 0.7 period of the pulse tube cycle?

Reply:

We used exactly the same data as obtained with the sequence shown in Fig 4a. Instead, we apply a different analysis from Figs. 4f and g (revised). As Referee #2 mentioned, we chunk the continuous monitoring data of the two qubits and see the occurrence of state flips from the initial states.

Regarding the second question, this analysis is done in a non-time-resolving manner, which gives the MI obtained from the data during the whole 0.7 period as Referee #2 stated.

Modification:

✓ We added the explanation on pages 16 of the marked-up main text.

7. I found that notations of “ 10^{-n} ” and “ e^{-n} ” are used interchangeably in the manuscript. I recommend sticking to one notation throughout the text.

Modification:

✓ We corrected them accordingly.

8. The authors claim that “We associate the mechanical sensitivity of the qubits with TLS- and quasiparticle-mediated interactions to their phononic baths” based on the analysis of the fluctuation of T1 overtime when the pulse tube is on. The fluctuation analysis can only prove that the qubit is TLS-limited similar to most highly coherent transmon qubits. From this analysis alone and statement such as “high-energy phonons break Cooper pairs”, I don’t think one can unambiguously associate the observation to TLS or quasiparticle.

Reply:

We fully agree that our experimental data cannot unambiguously associate the observations with TLS or quasiparticle dynamics. Nevertheless, we could exclude possible electrical mechanics that could be induced by side effects of the pulse tube and electric hammer operations, as well as the possibility of local heating or triboelectric effect. Therefore, we believe that there is a microscopic origin of the mechanical sensitivity of superconducting qubits. To stimulate future works triggered by our results, we suggest a possible origin that can be consistent with our observation using the TLS and quasiparticle models only in the introduction and discussion sections.

Modification:

✓ We removed the statement about TLSs and quasiparticles from the abstract.

Referee #3:

Kono et al. fabricate superconducting qubits with Nb electrodes that have relaxation times of several 100 microseconds. They perform time-resolved measurements of the population of the ground, first and second excited states of these qubits. They additionally characterise the vibrations of their dilution refrigerator caused by the pulse tube cooling unit by means of an accelerometer attached to the 300K plate of the instrument. The authors find a positive correlation between high vibrational noise and high excited state population as well as increased state switching rates. They perform control experiments to exclude electric and triboelectric noise (the latter only at a single frequency) and state that their observations would be compatible with TLS and quasiparticle-mediated qubit decay to a phononic bath.

I find the manuscript well-written, technically sound, and original. The data is thoroughly analysed and relevant control experiments are included. That being said, before recommending publication, I would like to see the following comments and questions addressed:

Reply:

We are grateful for Referee #3's positive assessment of our work. We carefully address their comments and questions. We believe that our manuscript has improved above the criteria for publication in Nature Communications.

1) Given that the authors state in the abstract that "Our observations are consistent with qubit dynamics induced by two-level systems and quasiparticles, thereby deepening our understanding of the qubit error mechanisms.", this point should be argued more conclusively in the manuscript. While the experiment is indeed original and interesting, there is no direct evidence indicating that TLSs and quasiparticles link the mechanical vibration and qubit dynamics. This statement is based on a rather speculative paragraph at the end of the discussion. I think this hypothesis can certainly be mentioned,

but should not be used as a central claim in the abstract. Therefore, the statement could be removed in an effort to not be misleading while, in my opinion, not detracting from the interest of the work.

Reply:

We agree with Referee #3 that we do not show direct evidence about the microscopic origin of the mechanical sensitivity of superconducting qubits.

Modification:

✓ We removed the statement about TLSs and quasiparticles from the abstract.

2) Y-axis of Fig.1 e and f as well as text: "excitation probability" could be read as "probability of exciting the qubit" as opposed to "probability of finding the qubit in the excited state" or in short: "excited-state probability". I believe the latter is what is meant here.

Modification:

✓ We modified them accordingly.

3) Fig2c: Could the authors comment on what the highest frequency harmonics extracted from the accelerometer data are?

Reply:

We provide spectral densities with a larger span in Supplementary Note 7, showing the harmonics are observed until approximately 50 Hz. Moreover, we found that there are higher-frequency noise components around 100 Hz and from 10 kHz to 100 kHz.

Modification:

✓ We added Supplementary Note 7 to show more detailed analysis of the acceleration data.

4) Fig2f,d and all other relevant plots: I suggest to normalise the quadrature axis to the standard deviation of the Gaussian fit corresponding to the qubit ground state. This adds more information compared to arbitrary units.

Modification:

✓ We modified all the relevant plots accordingly.

✓ We added the explanation about the data processing in the caption of Fig. 2.

5) "readout quadrature outcomes of approximately 3×10^5 and 10^7 when...": missing word: "measurements". Please justify why fewer measurements were taken in the "off" state than in the "on" state.

Reply:

The reason why the smaller number of data points with the pulse tube off is the measurement time without the pulse tube was limited to less than 5 min every try due to nonnegligible temperature increase at the 4-K and 50-K stages. After each try, we, therefore, needed to condense the collected mixture and wait for about one hour until the temperatures of all the stages were stabilized.

Nevertheless, we believe that the amount of data points is sufficiently large to see the difference in correlated bit-flip errors with and without the pulse tube operation.

Modification:

✓ We corrected the sentence accordingly.

✓ We added the explanation on page 10 of the marked-up main text.

6) Fig3a: The number of measurements is given in the text, but given the normalisation of the colour scale that information should also be in the figure caption.

Modification:

✓ We added the information also in the caption of Fig. 3.

7) Fig3 caption: typo: "(red shaded)" is given twice.

Modification:

✓ We corrected it.

8) "accumulate a sufficient number of the single-shot outcomes at an arbitrary time of interest within

the vibrational period to obtain the time-resolved residual excitation probabilities of the qubits." Please comment on what sets that sufficient number and add a one-standard deviation confidence interval on the time-resolved mutual information plotted in Figure 3e.

Reply:

"Sufficient number" simply means that the number of data points gives smaller statistical errors than the variation of the excited-state probabilities we observed (from 0.1 to 10 %).

Modification:

✓ We added the confidence intervals of one standard error to Fig. 3e.

9) "The time-resolved MI (blue solid line) is always smaller than the non time-resolved value (orange dashed line), implying the time-resolved events are not strongly correlated, but macroscopic parameters, such as the intrinsic decay rates or bath occupations, are changing synchronously for both the qubits, causing the correlated errors in the non-time resolved analysis." Could the authors elaborate on this further? How does this difference in time-resolved and non-time-resolved MI change with the timespan chosen for the time bin of the non-time-resolved MI?

Reply:

We have addressed the concern to answer Comment 5 from Referee 2 (see our answer there). Accordingly, when the analyzing time window is longer than the time scale of the variation of the qubit excited-state population, the single-shot readout data shows a certain correlation. On the other hand, the analyzing window shorter than the time scale of the variation does not show a strong correlation between the qubit excitation events (see the data of the time-resolved MI in Fig. 3e). For simplicity, we showed the two extreme cases: the shorter analyzing time span (1 ms) than the time variation (time-resolved data) and the longest time span (~ 0.7 s), i.e., one period of the pulse tube operation (non-time-resolved data).

Modification:

✓ We modified the explanation to be clear.

10) Section: "Mechanically-induced correlated jumps": In this section is the F-state population added to the E-state population or neglected? Please comment on how this choice influences the results in particular the error matrix Fig5d.

Reply:

We thank Referee #3 for the useful suggestion, which allows us to explore a new aspect of the qubit nonequilibrium dynamics induced by mechanical shocks. Indeed, in the original manuscript, we do not distinguish between G and E, leading to the F-state contribution added to the population of E. In the revised manuscript, we distinguished between the three states using the threshold in the complex plane for the readout outcomes (see Fig. S5a). We applied the new state-discrimination method for the analyses for Fig. 3d, Fig. 4, Fig. 6d, Fig. S5, Fig. S6, Fig. S7, and Fig. S8. As a result we find that the effects of the F state on the analyses are not negligible especially when the pulse tube mechanical shocks are dominant. This strongly supports that the qubits are not simply excited by thermal electrical noise or local heating, but rather suffer from nonequilibrium dynamics induced by mechanical shocks in the revised manuscript.

Modification:

- ✓ We modified Fig. 3d, Fig. 6d, and Fig. S5.
- ✓ We added Fig. 4, Fig. S6, Fig. S7, and Fig. S8.
- ✓ We added Supplementary Note 4.

11) Fig5 caption: "The shaded regions in (a) and (b) depict the fitting errors." What exactly is meant by fitting errors here? Could you formulate this in terms of residual standard deviation?

Reply:

We fitted exponential functions to the dwell time histograms. These analyses extracted the fitting parameters, i.e., the transition rates, as well as their fitting errors, which are shown in the figures.

12) "we can characterise the microwave transmission ratio and background noise at a time of interest

within the period of the vibrational noise". Please elaborate further on how exactly the background noise is extracted. In particular, how is the y-axis of Fig7b calibrated in units of noise quanta?

Reply:

Any linear (quadrature amplitude) detector can be equivalent to a single quadrature measurement with a quantum efficiency of η . Let's suppose we have a linear detector with a measurement scaling factor of C_m and an input-referred photon noise of n_m . The outcome, which is normalized by the outcome for the vacuum input and rescaled to 0.5, corresponds to the outcome obtained with a quadrature detector with an efficiency of $\eta = 0.5/(0.5 + n_m)$. In our analysis, we normalized the outcome by the non-time-resolved (averaged) noise, corresponding to the possible lowest noise since there is no significant increase in the time-resolved noise data. This is valid as long as the measurement environment is close to being in the vacuum, at least when the pulse tube effect is minimal. A finite thermal photon number would affect the estimated quantum efficiency, which we can safely assume is quite small given the heavily attenuated, filtered, and isolated input and output lines.

While we assumed in the first draft that the quantum efficiency is in the order of 0.1 from the literature, we estimated the quantum efficiency of our quadrature measurement in the revised manuscript. First, we calibrated the attenuation in the fridge by using photon-number-resolved qubit spectra. With this, we estimated the readout photon flux incoming to the readout cavity, resulting in the readout-induced qubit dephasing rate. By taking the ratio of the signal-to-noise rate of the single-shot readout to the inferred dephasing rate, we characterized the quantum efficiency of our measurement chain as $\eta > 0.2$. This is sufficient to conclude possible electrical noise in the qubit microwave environment cannot play a dominant role in exciting the qubits.

Modification:

- ✓ We added a new section to the method section of the main text.
- ✓ We added Supplementary Note 5.

13) Fig7b: this figure is very busy and pixelated when zooming in. Please provide the data in higher resolution.

Modification:

- ✓ We broke the original Fig. 7b into three different figures (b,c,d) with a 300-dpi resolution.

14) Fig8: Is there indeed a Cryoperm shield thermalised to the 50K plate and a Cu shield surrounding the entire 10mK stage setup? If so, please comment on the reasons for this. What about shielding thermalised to 3K and 0.8K?

Reply:

The outer Amumetal shield provided by BlueFors is equipped and thermalized to the vacuum can (300 K) while a Cu radiation shield is thermalized to the 10-mK stage, covering the entire 10 mK setup. We also have 50-K, 3-K and 0.8-K radiation shields as usual. In addition to the "standard" setup defined by BlueFors, we have homemade tri-layer shields: Amumetal 4K, Al, and Cu, to further protect qubits from electric and magnetic noise. We haven't checked if this heavily shielded setup is necessary to realize our highly coherent qubit. A careful investigation will be an interesting future work to understand the optimal experimental setup for such highly coherent qubits.

Modification:

- ✓ We modified Fig. 1d and Fig. 8 to be clear.
- ✓ We added the explanation to the method section of the main text.

REVIEWERS' COMMENTS

Reviewer #1 (Remarks to the Author):

I am generally happy with the replies to my prior remarks and with the corresponding changes to the manuscript. I also appreciated the constructive dialog with the other two reviewers and the corresponding improvements to the text.

Before publication I recommend addressing the following two points which arose following the revision:

1) In the revised version the dwell time histograms have been relegated from main text Fig. 4 to supplementary figures S5 and S6. I think this is not a wise decision, because dwell time histograms contain much more information about qubit dynamics than a single number (i.e. the decay rate). In fact, I find it interesting that the measured histograms are strongly deviating from exponential decays in both cases: pulse tube ON or OFF. Since this anomalous behavior is indicative of other mechanisms at play during qubit population relaxation (for a recent example see Spiecker et al. Two-level system hyperpolarization using a quantum Szilard engine, Nat. Phys. 19 (2023)), in my opinion it is of high importance to report it in the main text. It is likely a central piece of information in elucidating the currently open problem of the origin of mechanical sensitivity of the qubits.

2) My second point is more technical and is related to the first one. Looking closely at the histograms in S5 and S6 in the updated manuscript, it is clear that most of the qubit population decay occurs in the first couple of bins (in some case in the first bin). However, the exponential fits exclude these first bins, as illustrated by the lines in plots S5 and S6. Since during the first few bins the decay rate is much higher, I find this selection problematic. Given that the corresponding fitted decays rates are used in the error budget for single-shot qubit readout, this choice of fitting sheds doubt on the stated values for readout fidelity. While this is not the central point of the manuscript, it is mentioned in the main text, therefore the evaluation should be solid.

Reviewer #2 (Remarks to the Author):

The authors have answered my questions satisfactorily. Although the exact origin of the coupling is not pinpointed from the work, the correlation established here is of great interest to the community and can inspire many future research. I recommend the publication of this work in nature communications.

Reviewer #3 (Remarks to the Author):

I would like to thank the authors for engaging constructively with my comments and addressing them in a satisfying way. I recommend the manuscript for publication in Nature Communications. I have two minor comments that the authors could address to further improve the quality of the manuscript:

1) Fig.2, Fig S5, and all other relevant plots: after carrying out the suggested normalisation, the axes of the histogram plots are not in arbitrary units any more. The "(a.u.)" could now either be removed or the axis label could be Q/σ_g (or similar). If the first option is chosen, it would be good to mention in the caption of Fig.2 that the same normalisation was carried out everywhere else in the work.

2) In the discussion of Fig.4 the authors discuss the transition process between the G and F states even observed in the "quiet" period. It would be interesting to also comment on the fact that the $e \rightarrow f$ transition rate is almost two orders of magnitude larger than the $g \rightarrow e$ transition rate in the quiet period.

Reply to Referee Comments

Manuscript “Mechanically induced correlated errors on superconducting qubits with relaxation times exceeding 0.4 milliseconds” (NCOMMS-23-30295A)

Below we repeat the Referees’ reports verbatim, with our reply in **blue** and actions taken in the manuscript in **red** bullets.

◆ Referee #1:

I am generally happy with the replies to my prior remarks and with the corresponding changes to the manuscript. I also appreciated the constructive dialog with the other two reviewers and the corresponding improvements to the text.

Reply:

We appreciate Referee #1’s positive feedback on our revised manuscript. We finalized the manuscript based on all the comments.

Before publication I recommend addressing the following two points which arose following the revision:

1) In the revised version the dwell time histograms have been relegated from main text Fig. 4 to supplementary figures S5 and S6. I think this is not a wise decision, because dwell time histograms contain much more information about qubit dynamics than a single number (i.e. the decay rate). In fact, I find it interesting that the measured histograms are strongly deviating from exponential decays in both cases: pulse tube ON or OFF. Since this anomalous behavior is indicative of other mechanisms at play during qubit population relaxation (for a recent example see Spiecker et al. Two-level system hyperpolarization using a quantum Szilard engine, Nat. Phys. 19 (2023)), in my opinion it is of high importance to report it in the main text. It is likely a central piece of information in elucidating the currently open problem of the origin of mechanical sensitivity of the qubits.

Reply:

We agree with Referee #1’s useful suggestion. Moreover, investigating the dwell-time distributions carefully, we found two characteristic timescales that exist in some parts of our dwell-time dataset. To characterize this behavior and estimate the transition rates more accurately, we compared two fitting models: a standard exponential distribution ($f(\tau) = \Gamma e^{-\Gamma\tau}$) and a mixture of two exponential distributions ($f(\tau) = p\Gamma_1 e^{-\Gamma_1\tau} + (1-p)\Gamma_2 e^{-\Gamma_2\tau}$) where we can extract the effective transition rate as a weighted average $\Gamma = p\Gamma_1 + (1-p)\Gamma_2$. Finally, we selected the preferable model based on the Bayesian Information Criterion for each dwell-time histogram analysis. Although the physical origin

of this behavior is not determined unambiguously in this work, similar double exponential behaviors are observed in different qubit systems, which could be explained by nonequilibrium quasiparticles. Moreover, we believe that the TLS saturation effect cannot simply explain our double exponential behavior since we have not polarized the two-level system bath unlike the experiments in Nat. Phys. 19 (2023). Nevertheless, we believe the new fitting method will represent the true transition rates more accurately.

Modifications:

- ✓ We modified Fig.4 and put back the G , E , and F dwell-time histograms when the mechanical shock perturbations are minimal and maximal.
- ✓ We updated Fig4, FigS5-7 with the new fitting method.
- ✓ We added the explanations in the main text and the SI, and make FigS6e-g to explain the new fitting method.

2) My second point is more technical and is related to the first one. Looking closely at the histograms in S5 and S6 in the updated manuscript, it is clear that most of the qubit population decay occurs in the first couple of bins (in some case in the first bin). However, the exponential fits exclude these first bins, as illustrated by the lines in plots S5 and S6. Since during the first few bins the decay rate is much higher, I find this selection problematic. Given that the corresponding fitted decays rates are used in the error budget for single-shot qubit readout, this choice of fitting sheds doubt on the stated values for readout fidelity. While this is not the central point of the manuscript, it is mentioned in the main text, therefore the evaluation should be solid.

Reply:

We greatly thank Referee #1 for their insightful comments. We numerically simulated the quantum jump events and obtained the dwell-time distributions by taking into account the readout separation and state-flip errors. While the dynamics associated with the state-flip errors correspond to what we want to measure, we found that the separation errors have the following two nontrivial effects on the distributions and the transition rates.

First, the readout separation errors induce fake jump events, where a readout process assigns the qubit state wrongly due to the separation errors and the following readout assigns the state correctly the most probably. This event shows up as a peak in the first bin of the dwell-time histograms (as Referee #1 mentioned). However, these contributions can be mitigated easily by neglecting the first bin for fitting, as we do.

Second, the readout separation errors effectively increase the transition rates. The probability that a correct readout outcome of state x transits to the other states in the following readout is given by $\Gamma_x \tau + \epsilon_x$, where Γ_x is the transition rate from state x to the others and τ is the total readout time, and ϵ_x is the readout separation error probability for state x . Therefore, the measured transition rates are effectively modified to be $\Gamma_x + \epsilon_x/\tau$. Nevertheless, since the readout separation errors are

independently characterized by the readout quadrature histograms, we can correct these modifications straightforwardly.

Finally, as forementioned, we improved the fitting method to capture two different characteristic timescales existing in some parts of our dwell-time dataset and updated the transition rates and readout error accordingly.

Modifications:

- ✓ We added explanations about the correction of the separation error contribution on the transition rates in the main text and the SI.
- ✓ We recalculated the transition rates taking into account the readout separation errors and updated Figs 4 and S7.

◆ **Referee #2:**

The authors have answered my questions satisfactorily. Although the exact origin of the coupling is not pinpointed from the work, the correlation established here is of great interest to the community and can inspire many future research. I recommend the publication of this work in nature communications.

Reply:

We thank Referee #2 for the positive comments and the agreement on the publication.

◆ **Referee #3:**

I would like to thank the authors for engaging constructively with my comments and addressing them in a satisfying way. I recommend the manuscript for publication in Nature Communications.

Reply:

We thank Referee #3 for the positive comments and the agreement on the publication. We finalized the manuscript accordingly.

I have two minor comments that the authors could address to further improve the quality of the manuscript:

- 1) Fig.2, Fig S5, and all other relevant plots: after carrying out the suggested normalisation, the axes of the histogram plots are not in arbitrary units any more. The "(a.u.)" could now either be removed or the axis label could be Q/σ_g (or similar). If the first option is chosen, it would be good to

mention in the caption of Fig.2 that the same normalisation was carried out everywhere else in the work.

Modification:

✓ We follow the first option that Referee #3 suggested and modified the manuscript accordingly.

2) In the discussion of Fig.4 the authors discuss the transition process between the G and F states even observed in the “quiet” period. It would be interesting to also comment on the fact that the $e \rightarrow f$ transition rate is almost two orders of magnitude larger than the $g \rightarrow e$ transition rate in the quiet period.

Reply:

This is an important point to mention. Upon the insightful suggestions raised by Referee #3, we considered the effect of the readout separation errors on the transition rates and found that the separation errors modify the transition rates since the errors effectively increase the transition probabilities. In the revised manuscript, we independently characterized the readout separation errors and subtracted the separation error contributions from the modified transition rates. It turns out that the transition rate from G to E is roughly comparable with or slightly smaller than that from E to F in the “quiet” period, which is reasonable due to the difference in the transition moments.

Modification:

✓ We recalculated the transition rates taking into account the readout separation errors and updated Figs 4 and S7.

◆ **Additional modifications to be mentioned:**

- ✓ We showed all the error probabilities in %.
- ✓ We add another scenario of local heating for the case study of mechanical shock side effects on the qubit excitations.
- ✓ We added references (25, 44, 55, 56, 57) to the revised main manuscript and (10) to the SI, respectively.
- ✓ We optimized the definitions of the thresholds for state discriminations.
- ✓ We used the continuous monitoring dataset to characterize the state discrimination thresholds and the readout separation errors for determining the transition rates using the dwell time events.
- ✓ We changed the color range in Fig. 5c for a better visualization.